# Single-molecule imaging and microfluidic platform reveal molecular mechanisms of leukemic cell rolling

Bader Al Alwan[1], Karmen AbuZineh[1], Shuho Nozue[1], Aigerim Rakhmatulina[1], Mansour Aldehaiman[1], Asma S. Al-Amoodi[1], Maged F. Serag[1], Fajr A. Aleisa[1], Jasmeen S. Merzaban[1,2✉] & Satoshi Habuchi[1,2✉]

Hematopoietic stem/progenitor cell (HSPC) and leukemic cell homing is an important biological phenomenon that occurs through key interactions between adhesion molecules. Tethering and rolling of the cells on endothelium, the crucial initial step of the adhesion cascade, is mediated by interactions between selectins expressed on endothelium to their ligands expressed on HSPCs/leukemic cells in flow. Although multiple factors that affect the rolling behavior of the cells have been identified, molecular mechanisms that enable the essential slow and stable cell rolling remain elusive. Here, using a microfluidics-based single-molecule live cell fluorescence imaging, we reveal that unique spatiotemporal dynamics of selectin ligands on the membrane tethers and slings, which are distinct from that on the cell body, play an essential role in the rolling of the cell. Our results suggest that the spatial confinement of the selectin ligands to the tethers and slings together with the rapid scanning of a large area by the selectin ligands, increases the efficiency of selectin-ligand interactions during cell rolling, resulting in slow and stable rolling of the cell on the selectins. Our findings provide novel insights and contribute significantly to the molecular-level understanding of the initial and essential step of the homing process.

[1] King Abdullah University of Science and Technology (KAUST), Biological and Environmental Science and Engineering Division, Thuwal, Saudi Arabia. [2] These authors contributed equally: Jasmeen S. Merzaban, Satoshi Habuchi. ✉email: jasmeen.merzaban@kaust.edu.sa; satoshi.habuchi@kaust.edu.sa

Delivering circulatory cells to specific sites in the body is central to many physiological functions, from immunity to cancer metastasis, which is achieved by their interactions with the surface of endothelium under the presence of external shear forces[1,2]. These cellular interactions are controlled by a number of adhesion molecules that include selectins and integrins and their corresponding ligands[3]. So far, molecular mechanisms of cell adhesion have been investigated by characterizing migration behavior at the cellular level in the presence of shear force exerted to the migrating cells and/or characterizing binding behavior of the adhesion molecules with their ligands at the molecular level by applying an external force to the bonds using, for example, single-molecule force spectroscopy technique[4–6]. However, under physiological flow conditions, interactions between adhesion molecules and their ligands occur under a spatiotemporal rather heterogeneous cellular environment. Thus, without capturing real-time nanoscopic spatiotemporally interactions of adhesion molecules and their ligands at the molecular level in the cellular environment, one cannot develop a complete picture of complicated cellular interactions that exist.

Hematopoietic stem/progenitor cell (HSPC) homing is an important biological phenomenon in which transplanted HSPCs travel from the peripheral blood to their home in the bone marrow[7]. The homing of HSPCs, as well as leukemic cells, to bone marrow is a multistep process that is initiated by the tethering and rolling of cells, expressing E-selectin ligands such as CD44[8,9] and PSGL-1[10,11], on endothelium expressing E-selectin[4,7,12–14]. Besides mediating their slow rolling[15] along the endothelium at shear stresses of several dynes cm$^{-2}$ generated by the blood flow[16–18], E-selectin may also regulate tumor growth and proliferation of leukemic cells within the bone marrow[19]. Therefore, molecular level understanding of E-selectin binding to its ligands is essential. Multiple factors that affect the rolling behavior of the cells have been identified, including spatial clustering of the ligands[20], formation of membrane tethers and slings[21], and shear force-dependent selectin-ligand interactions[6], spatiotemporal dynamics of selectin ligands during this initial step of homing and its contribution to slow and stable cell rolling are not well understood. This is due to the lack of an experimental method that enables capturing real-time nanoscopic spatiotemporal interactions of adhesion molecules and their ligands at the molecular level in the cellular environment.

In this study, using a microfluidics-based single-molecule live-cell fluorescence imaging technique, we showed that the unique spatiotemporal dynamics of selectin ligands on the membrane tethers and slings play an essential role in the rolling of the cell. We demonstrated that the membrane tethers are formed from single microvilli on the cells and this provides a mechanism to spatially localize selectin ligands, in particular PSGL-1 and CD44, on these tethers and slings. Furthermore, this work also established that due to the detachment of the cell from the actin cytoskeleton during the formation of the tethers, the fast and random diffusional motion of selectin ligands is exhibited along these structures (i.e., tethers and slings) in contrast to the slow and confined motion of the ligands on the cell body. Our results suggest that the spatial confinement of the selectin ligands to the tethers and slings together with the rapid scanning of a large area by the selectin ligands increases the efficiency of selectin-ligand interactions during cell rolling, resulting in slow and stable rolling of the cell on the selectins.

## Results and discussion

**Experimental design.** We developed a microfluidics-based single-molecule fluorescence imaging platform to capture and characterize molecular level spatiotemporal dynamics of selectin ligands that occur during cell rolling on selectins[22]. In this study, we used this imaging platform to characterize the physical mechanisms of the formation of membrane tethers and slings that extend backward and forward of the rolling cells. To this end, we deposited recombinant human (rh) E-selectin on the surface of a microfluidic chamber (Supplementary Fig. 1) at the surface density of 0.6–111 molecules μm$^{-2}$. A suspension of primary human CD34$^{pos}$-HSPCs or KG1a cells, a human leukemic progenitor cell line often used as a model system of HSPCs[23–25], were injected into the fluidic chamber at the shear stresses of 1–8 dyne cm$^{-2}$ (0.1–0.8 Pa) using a syringe pump (Fig. 1). The number of KG1a cells bound to the surface E-selectin increased with the increase of the surface density of E-selectin (Supplementary Fig. 2a), whereas the rolling velocity of the KG1a cells decreased with the increase of the surface density (Supplementary Fig. 2b). We observed that the binding of the KG1a cells to E-selectin requires calcium ion (Supplementary Fig. 2c). All these observations suggest that the tethering and rolling of the cells are caused by the specific interaction between the selectin ligands on the cells and the surface E-selectin, mimicking in vivo cell rolling behavior. The surface density of E-selectin was set to 15 molecules μm$^{-2}$ unless otherwise described, which is close to the number density of E-selectin on endothelial cells[26,27].

The cytoplasmic membrane of the primary human CD34$^{pos}$-HSPCs and KG1a cells was fluorescently stained using Vybrant DiO dye. CD44 and PSGL-1, two selectin ligands expressed on these cells were immunolabeled by fluorophore (either Alexa-Fluor-488, Alexa-Fluor-555, or Alexa-Fluor-647)-conjugated antibodies (515 mAB and KPL-1 mAB for CD44 and PSGL-1, respectively) and introduced into the chamber. Fluorescence-activated cell sorting (Supplementary Figs. 3 and 4) and fluorescence imaging of labeled cells (Supplementary Fig. 5) were used to confirm the binding specificity of these antibodies. Since antibodies have two binding sites to their epitopes that may cause artificial clustering of the labeled molecules, we immunostained PSGL-1 on KG1a cells using the Alexa-Fluor-555-conjugated Fab fragment of the anti-PSGL-1 antibody (Alexa-555-Fab) that has only one binding site to its epitope. The immunofluorescence image of PSGL-1 on the membrane tethers and slings of the KG1a cells (see below) obtained using the Alexa-555-Fab showed a spatial distribution of the PSGL-1 molecules very similar to that obtained using the Alexa-Fluor-555-conjugated anti-PSGL-1 antibody (Supplementary Note 1, Supplementary Fig. 6). This result demonstrates that the spatial distribution of the selectin ligands on the cells are not the result of artificial clustering of

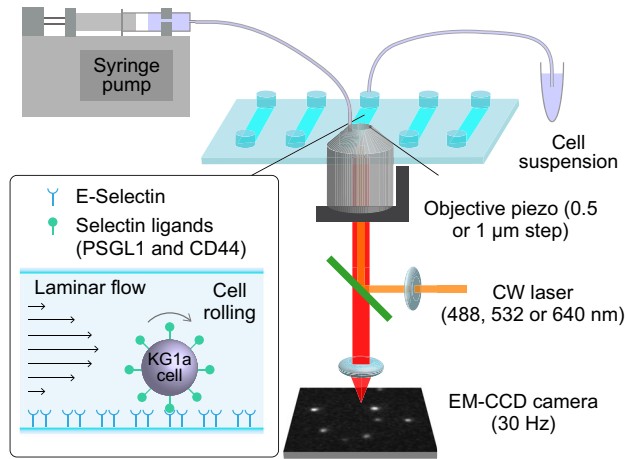

**Fig. 1 Experimental configuration.** Schematic illustration describing the custom-built microfluidics-based single-molecule imaging setup.

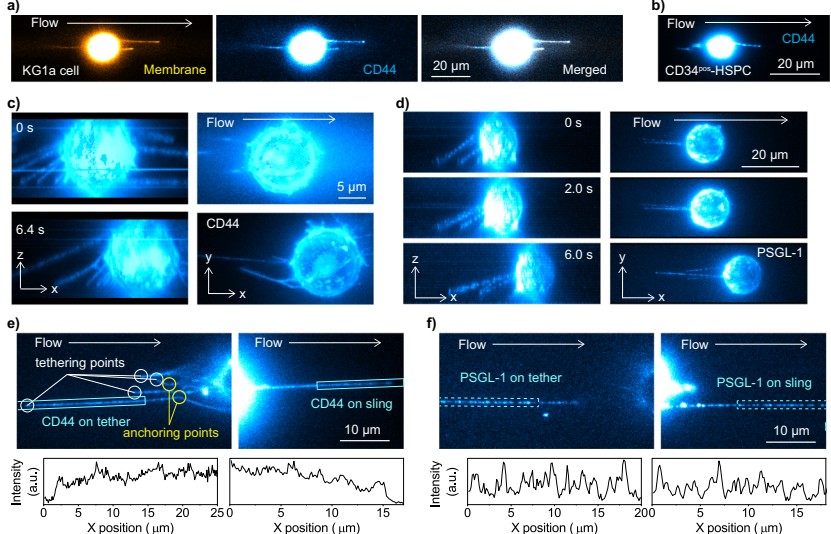

**Fig. 2 Single-molecule visualization of selectin ligands on the tethers and slings. a** Two-color fluorescence images of the cell membrane (left, stained by Vybrant DiO dye) and CD44 (center, immunostained by Alexa-Fluor-647-conjugated anti-CD44 antibody) captured during the KG1a cell rolling over E-selectin. The right panel shows the merged image. **b** An example of the fluorescence images of CD44 (immunostained by Alexa-Fluor-647-conjugated anti-CD44 antibody) captured during human CD34$^{pos}$-HSPC rolling over E-selectin. **c, d** 3D views of the tethers and slings formed on a KG1a cell rolling over E-selectin by (**c**) CD44 and (**d**) PSGL-1. Side views (left) and top views (right) of the 3D reconstructed time-lapse fluorescence images of CD44 (immunostained by Alexa-Fluor-647-conjugated anti-CD44 antibody) and PSGL-1 (immunostained by Alexa-Fluor-555-conjugated anti-PSGL-1 antibody) captured during cell rolling over E-selectin. The 3D images were reconstructed by recording fluorescence images of the cell at 53 different Z-axis positions with 0.5 μm step size and 33 different Z-axis positions with 1.0 μm step size for CD44 and PSGL-1, respectively. All the fluorescence images of the rolling cells were captured by injecting KG1a cells into the rh E-selectin-deposited microfluidic chambers at a shear stress of 2 dyne cm$^{-2}$ (0.2 Pa). **e, f** Immunofluorescence image of the (**e**) CD44 and (**f**) PSGL-1 molecules on the tethers (top left) and slings (top right) formed during cell rolling over E-selectin. The bottom left and bottom-right panels show fluorescence intensity profiles along the tethers and slings obtained from cyan regions in the top left and top right panels, respectively. All the fluorescence images of the rolling cells were captured by injecting KG1a cells into the rh E-selectin-deposited microfluidic chambers at a shear stress of 2 dyne cm$^{-2}$ (0.2 Pa).

PSGL-1 due to the bivalency effect of the antibody. We also confirmed that the pretreatment of KG1a cells with antibodies to PSGL-1 or CD44 did not affect their ability to bind E-selectin (Supplementary Fig. 7, Supplementary Note 2).

**Visualization of membrane tethers/slings and selectin ligands.** Standard 2D epi-fluorescence microscopy images of the Vybrant DiO-stained KG1a cells rolling on E-selectin at a shear stress of 2 dyne cm$^{-2}$ (0.2 Pa) illustrated the formation of membrane tethers and slings (Fig. 2a, Supplementary Fig. 8)[21]. Membrane tethers that appear during cell rolling on selectins are believed to play a critical role in the stable rolling of the cells. Previous studies demonstrated that live cells roll on a selectin surface more stably compared to fixed cells or to microspheres coated with selectin ligands[14]. These observations suggest that tethers help decrease the tension exerted on selectin-ligand bonds and thereby reduce the probability of breaking these interactions. This stronger binding is consistent with previous studies from our lab illustrating that the off rate of binding between selectin ligands and E-selectin is low[23,25]. Two-color epi-fluorescence imaging of the cell membrane and CD44 of the KG1a cells showed perfect colocalization of the two images, demonstrating that the tethers and slings can also be visualized by the immunofluorescence image of the CD44 molecules (Fig. 2a, Supplementary Note 3, Supplementary Fig. 9). Using the immunofluorescence images of CD44, we confirmed that the membrane tethers and slings are formed on the CD34$^{pos}$-HSPCs during the cell rolling in a way similar to those formed on the KG1a cells (Fig. 2b).

To fully visualize the formation of the tethers and slings during cell rolling along with the spatial distributions of the selectin ligands on the tethers and slings, we reconstructed 3D fluorescence images of CD44 and PSGL-1 on the rolling KG1a cells by recording epi-fluorescence images of the specimens at different Z-axis positions with 0.5–1.0 μm step size (Fig. 1). Due to the fluorescence from out-of-focus regions, the reconstructed 3D fluorescence images of the cell body are distorted along the Z-axis and fine structures on the cell body cannot be resolved clearly. On the other hand, the width of the membrane tethers and slings formed during the cell rolling (see below) captured by the fluorescence images of CD44 and PSGL-1 is close to the diffraction-limited size defined by the Rayleigh criterion (0.21–0.33 μm in our experimental conditions). The depth of field (Z) in our experimental conditions is calculated by using the following equation[28]

$$Z = \frac{n\lambda}{NA^2},\qquad(1)$$

where $n$ is the refractive index of the medium between the objective lens and the specimen. $\lambda$ and NA denote the wavelength of light (i.e., fluorescence wavelength) and the numerical aperture of the objective lens. Given the optical characteristics of the objective lenses ($n = 1.51$, NA = 1.49 for UAPON 100XOTIRF and $n = 1.41$, NA = 1.25 for UPLSAPO40XS) and the spectroscopic properties of the fluorophores ($\lambda = 0.5$–0.67 μm) used in this study, the depth of field in our experiments is estimated to be in the range of 0.34–0.60 μm. Since the physical size (i.e., width and depth) of the tethers and slings is smaller than the depth of field, our imaging mode enables us to reconstruct the 3D image of the tethers and slings at the spatial resolution defined by the depth of field without the effect of out-of-focus fluorescence (Fig. 2c, d, Supplementary Figs. 10 and 11). 2D fluorescence images were captured using standard epi-illumination wide-field fluorescence microscopy. The spatiotemporal dynamics of the selectin ligands

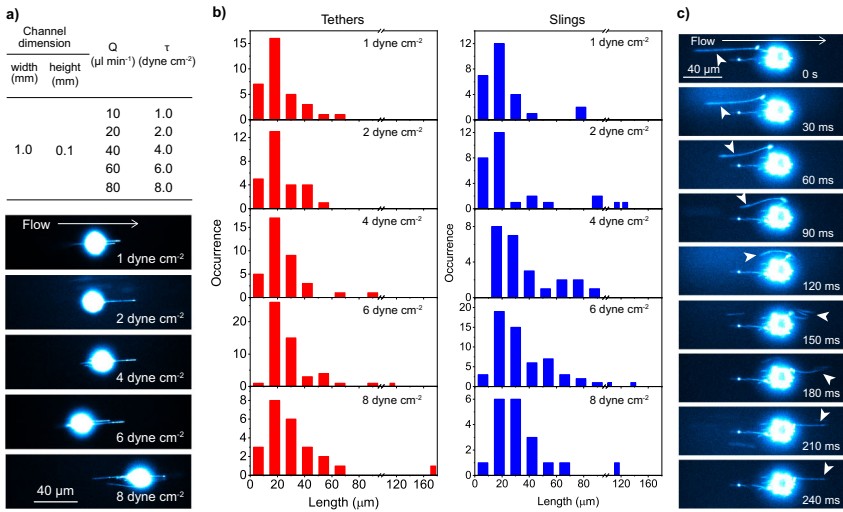

**Fig. 3 Spatiotemporal dynamics of the tethers and slings during cell rolling. a** Shear force-dependent formation of the tethers and slings on KG1a cells rolling over E-selectin. Fluorescence images of CD44 (immunostained by Alexa-Fluor-647-conjugated anti-CD44 antibody) captured during cell rolling over the surface-deposited E-selectin molecules. The cells were injected into the chambers at a shear stress of 1, 2, 4 or 8 dyne cm$^{-2}$ (0.1, 0.2, 0.4, and 0.8 Pa). The dimensions of the fluidic chamber, volumetric flow rates ($\tau$), and corresponding shear stress (Q) are inserted in the top. **b** Length of the tethers and slings formed on KG1a cells rolling over E-selectin. Frequency histograms of the length of tethers (red bars) and slings (blue bars) formed during the KG1a cells rolling over E-selectin. **c** Time-lapse fluorescence images of CD44 (immunostained by Alexa-Fluor-647-conjugated anti-CD44 antibody) captured during cell rolling over E-selectin. The arrow heads show a tether that is converted into a sling upon the detachment of the tethering point from the E-selectin surface. The cells were injected into the chambers at a shear stress of 6 dyne cm$^{-2}$ (0.6 Pa). All the fluorescence images of the rolling cells were captured by injecting KG1a cells into the rh E-selectin-deposited microfluidic chambers.

on the tethers and slings were captured at the single-molecule level by either 2D (Fig. 2e, f) or 3D epi-illumination wide-field fluorescence microscopy (Fig. 2c, d) similar to the visualization of the tethers and slings.

**Spatiotemporal dynamics of tethers and slings.** The tethers and slings were formed on KG1a cells at all the shear stress used in this study (1–8 dyne cm$^{-2}$ (0.1–0.8 Pa), Fig. 3a, Supplementary Fig. 12, Supplementary Movie 1). The formation of the tethers and slings was also observed for primary human CD34$^{pos}$-HSPCs at a shear stress of 2 dyne cm$^{-2}$ (0.2 Pa) (Fig. 2b, Supplementary Fig. 13). These are in contrast to the previous study on neutrophils rolling over P-selectin, which showed the formation of the tethers and slings only at shear stresses higher than 6 dyne cm$^{-2}$ (0.6 Pa) (Supplementary Note 4)[21]. The length of the tethers and slings range in size between several micrometers to tens of micrometers (Fig. 3b, Supplementary Fig. 14). The mean lengths of the tether and sling increased from 18 to 30 μm and from 13.5 to 30 μm for the tethers and slings, respectively when the applied shear stress was increased from 1 to 8 dyne cm$^{-2}$ (0.1–0.8 Pa) (Fig. 3b). Interestingly, on occasion, tethers and slings longer than 100 micrometers were observed (Fig. 3b). Time-lapse fluorescence images of CD44 clearly demonstrated that a tether detached from the E-selectin surface is converted into a sling within several hundreds of milliseconds in both KG1a cells and primary human CD34$^{pos}$-HSPCs (Fig. 3c, Supplementary Fig. 15, Supplementary Movie 2, Supplementary Note 5)[29]. Given that the primary human CD34$^{pos}$-HSPCs showed very similar structures to our model CD34$^{pos}$ cell line (i.e., KG1a cells) and that access to these primary cells is limited, we chose to focus on KG1a cells for all subsequent experiments. While the slings are persistent structures, we also observed a retraction of the slings during cell rolling (Supplementary Fig. 16, Supplementary Movie 3). The side views of the 3D images visibly show the formation of multiple tethers and slings during cell rolling over E-selectin (Fig. 2c, Supplementary

Fig. 10). In addition, the time-lapse 3D images captured the elongation of tethers (Supplementary Fig. 17) and the change in the Z-axis position of the slings (Supplementary Fig. 18) during cell rolling. We also often observed the formation of anchoring points on the tethers (Fig. 2e, Supplementary Fig. 19). The tether and sling dynamics including the elongation, anchoring, long lengths, and multiplicity of these structures are phenomena that have not been described previously and suggest mechanisms that are used by the cells that lead to the slow rolling of cells at a variety of shear stresses.

**Spatial clustering of selectin ligands at tethering and anchoring points.** The immunofluorescence images of CD44 on KG1a cells showed the contiguous distribution of CD44 on the tethers and slings (Fig. 2c, e). In contrast, we found that PSGL-1 shows a discrete spatial distribution on the tethers and slings (Fig. 2d, f, Supplementary Movies 4 and 5). The PSGL-1 molecules on the tethers and slings displayed perfect spatial overlap with CD44 and the cell membrane (Fig. 4a, b, Supplementary Figs. 20 and 21). The side views of the 3D images also unambiguously demonstrated that the tethers and slings are entirely covered with the CD44 and PSGL-1 molecules in a distinct spatial distribution (i.e., contiguous and discrete distributions for CD44 and PSGL-1, respectively, Fig. 2c, d). These results demonstrated that the entire tethers were consistently covered by the selectin ligands during their elongation (Fig. 2c, d, Supplementary Figs. 10 and 11).

The fluorescence images of CD44 and PSGL-1 on the tethers and slings sometimes show bright spots at their tips (Fig. 4c, d, and Supplementary Fig. 22), indicating clustering of the selectin ligands at the tethering points. The clustering of selectin ligands at the tethering points was further investigated by calculating the number of PSGL-1 molecules at each spot on the tethers and slings. Our calculation demonstrated that each PSGL-1 spot distributed on the tethers and slings mainly corresponds to a single PSGL-1 molecule (Fig. 4e). The distributions can be fitted

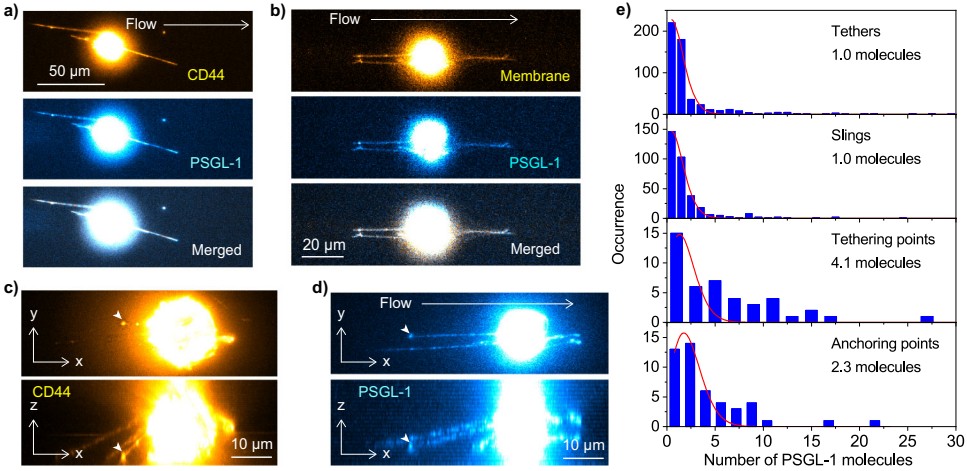

**Fig. 4 Spatial clustering of the selectin ligands at the tethering and anchoring points. a** Two-color fluorescence image of CD44 (top, yellow, immunostained by Alexa-Fluor-647-conjugated anti-CD44 antibody) and PSGL-1 (middle, cyan, immunostained by Alexa-Fluor-488-conjugated anti-PSGL-1 antibody) captured during the KG1a cell rolling over E-selectin. The merged image is displayed in the bottom panel. **b** Two-color fluorescence images of the cell membrane (top, yellow, stained by Vybrant DiO dye) and PSGL-1 (middle, cyan, immunostained by Alexa-Fluor-647-conjugated anti-PSGL-1 antibody,) captured during the KG1a cell rolling over the surface-deposited E-selectin molecules. The merged image is displayed in the bottom panel. **c**, **d** Top view (top) and side view (bottom) of the 3D reconstructed fluorescence images of the (**c**) CD44 molecules (immunostained by Alexa-Fluor-647-conjugated anti-CD44 antibody) and (**d**) PSGL-1 molecules (immunostained by Alexa-Fluor-555-conjugated anti-PSGL-1 antibody) captured during the KG1a cells rolling over E-selectin. The 3D images were reconstructed by recording fluorescence images of the cell at 59 different Z-axis positions with 0.5 μm step size and 40 different Z-axis positions with 1 μm step size for CD44 and PSGL-1, respectively. **e** Frequency histograms of the number of the PSGL-1 molecules in each spot on the tethers and slings and at the tethering points and anchoring points of the KG1a cells. The solid lines show Poissonian fittings. All the fluorescence images of the rolling cells were captured by injecting KG1a cells into the rh E-selectin-deposited microfluidic chambers at a shear stress of 2 dyne cm$^{-2}$ (0.2 Pa).

to Poisson distribution using the following equation

$$y = e^{-r} \frac{r^x}{x!}, \tag{2}$$

where $r$ is the mean of the distribution (Fig. 4e). This result indicates that the number of PSGL-1 molecules in each spot on the tethers and slings is determined in a stochastic way with the mean number of one PSGL-1 molecule per spot. Since the fluorescence labeling of the antibodies by the Alexa-Fluor-dyes also has a stochastic nature (i.e., each antibody carries a different number of the Alexa-Fluor-dyes whose distribution is described by Poisson distribution), the obtained Poisson distribution of the number of PSGL-1 molecules in each spot on the tethers and slings could be affected by this. However, the degree of labeling is in the range of 4–8 dyes per antibody, which should not give the experimentally observed Poisson distribution (i.e., mean number of one). Given the one-to-one binding of the antibody and PSGL-1, the result rather suggests that the number of PSGL-1 molecules in each spot on the tethers and slings is indeed determined in a stochastic way (i.e., absence of a mechanism that colocalizes the PSGL-1 molecules together) with the mean number of one PSGL-1 molecule per spot.

In contrast to the distributions of PSGL-1 on the tethers and slings, we found that multiple PSGL-1 molecules are present at the tethering point (4.1 molecules, Fig. 4e) and anchoring point (2.3 molecules, Fig. 4e), suggesting the spatial clustering of the selectin ligands at the tethering and anchoring points. The frequency histograms of the number of the PSGL-1 molecules in each spot at the tethering points (Fig. 4e) and anchoring points (Fig. 4e) show a large deviation from Poisson distribution. The result suggests that the number of PSGL-1 molecules in each tethering and anchoring point is not determined in a stochastic way. Instead, the result suggests the presence of a specific mechanism that supports the spatial clustering of PSGL-1 in the tethering and anchoring points. The most likely interpretation of

the observation is that the formation of the tethering points and anchoring points is facilitated by the binding of multiple PSGL-1 molecules located in the close vicinity (i.e. spatially clustered PSGL-1) to the surface E-selectin. The fact that larger numbers of the PSGL-1 molecules exist at the tethering points compared with those at anchoring points supports this interpretation. Since the anchoring points are formed after the initial tethering events occur, the tension exerted to the anchoring points should be smaller than that exerted to the tethering points. Thus, the smaller numbers of PSGL-1 at the anchoring points is an indication that the initial tether formation requires the binding of multiple PSGL-1 molecules to the surface E-selectins.

**Spatial clustering of selectin ligands supports the tethers/slings formation**. To investigate the effect of the clustering of the selectin ligands on the formation of tethers/slings and on the rolling behavior of the cell, we disrupted the clusters of CD44 by treating the cells with methyl-β-cyclodextrin (MβCD)[20,22,30]. Cholesterol existing in the cell membrane is extracted by this treatment, leading to the disruption of lipid rafts domains. Super-resolution fluorescence localization microscopy images of CD44 on the KG1a cells clearly showed that the 200 nm-size CD44 clusters existing on the control cells are disrupted by the MβCD treatment (Fig. 5a, b)[22]. Scanning electron microscopy (SEM) images of the control and MβCD-treated KG1a cells clearly revealed that microvilli existing on the control cells (Fig. 5c, Supplementary Fig. 23a) disappear upon the MβCD treatment (Fig. 5d, Supplementary Fig. 23b). These results are reminiscent of previous data from our lab showing that CD34 plays a key role in the formation of microvilli structures since once they are knocked down, microvilli structures disappear[25]. This result strongly suggests that the CD44 molecules localize on the microvilli in the control cells.

We found that the formation of the tethers and slings is extremely inefficient under the MβCD-treated condition (Fig. 5e,

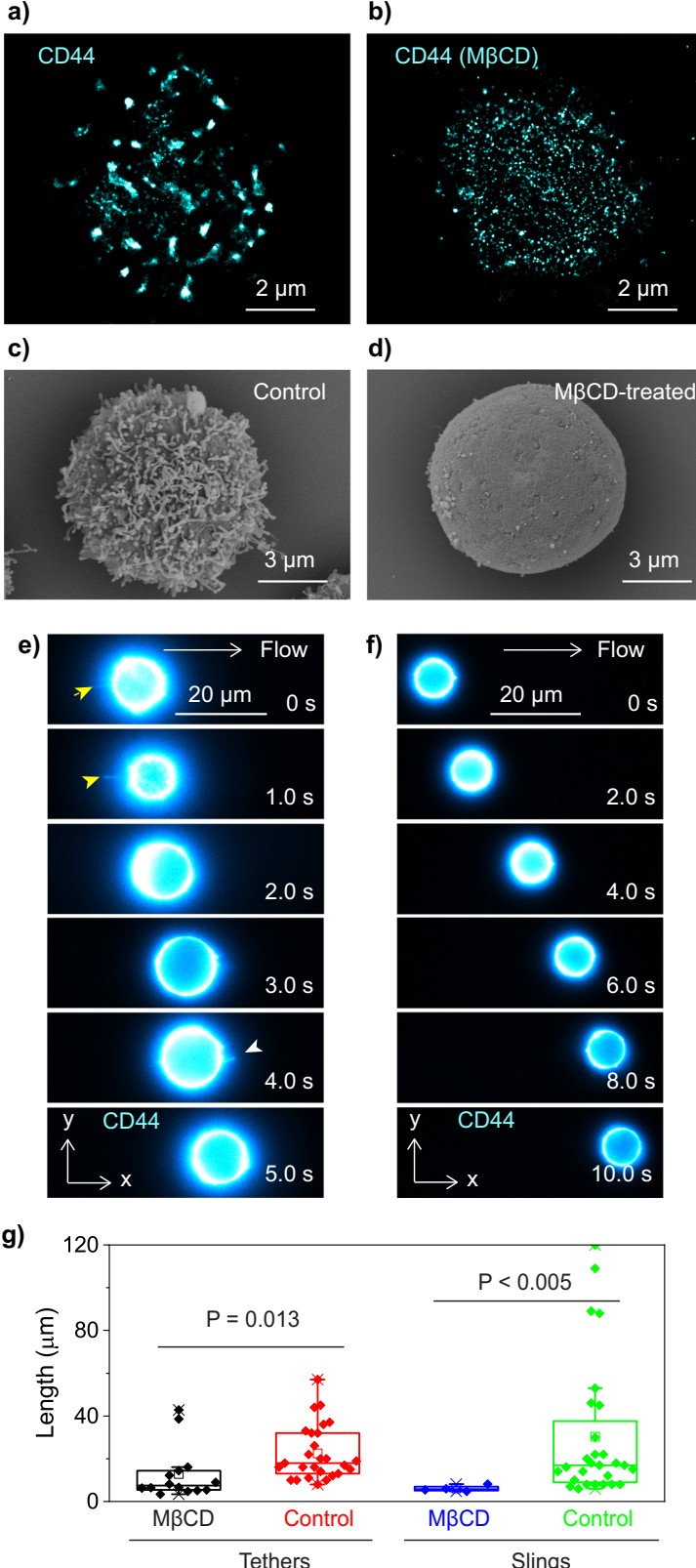

**Fig. 5 Effect of microvilli on the formation of the tethers and slings. a, b** Super-resolution fluorescence localization microscopy image of CD44 (immunostained by anti-CD44 515 antibodies followed by Alexa-Fluor-647-conjugated secondary antibodies) on the (**a**) control KG1a cell and (**b**) methyl-β-cyclodextrin (MβCD)-treated KG1a cell. **c, d** Scanning electron microscopy image of the (**c**) control KG1a cell and (**d**) MβCD-treated KG1a cells. **e, f** Time-lapse fluorescence images of the CD44 molecules (immunostained by Alexa-Fluor-647-conjugated anti-CD44 antibody) captured during the (**e**) MβCD-treated KG1a cell and (**f**) CD34 knockdown (CD34-KD) KG1a cell rolling over E-selectin. The yellow and white arrowheads highlight tether and sling, respectively. **g** Box plots showing the length of individual tethers and slings on the control and MβCD-treated KG1a cells. All the fluorescence images of the rolling cells were captured by injecting KG1a cells into the rh E-selectin-deposited microfluidic chambers at a shear stress of 2 dyne cm$^{-2}$ (0.2 Pa).

Supplementary Note 6, Supplementary Fig. 24) as well as when CD34 is knocked down (Fig. 5f, Supplementary Figs. 25 and 26). While the tethers and slings are formed in more than 90% of the control cells rolling over E-selectin (Supplementary Movie 6), they are observed in <10% of the MβCD-treated cells and CD34 knockdown cells. Further, the length of the tethers and slings formed during the rolling of the MβCD-treated cells were much shorter than those formed in the control cells (Fig. 5g). The fluorescence images of CD44 on the MβCD-treated cells and CD34 knockdown cells never showed bright spots (i.e., clusters of CD44) at the tethering points (Fig. 5e, f). These results demonstrate that the clustering of the selectin ligands contributes to both initial bindings to the surface E-selectin (i.e., tether formation) as well as the elongation of the tethers against the tension exerted to the tethering point during cell rolling.

Since CD34 knockdown KG1a cells and MβCD-treated KG1a cells exhibit ~3–5 times faster rolling velocity compared with control cells and showed more unstable rolling (i.e., faster detachment) on E-selectin[22,25], our findings suggest that the efficient formation of the tethers and slings and their strong resistance to shear stress due to the spatial clustering of the selectin ligands on the microvilli enables the slow and stable rolling of the cells. Although it is still controversial[31,32], PSGL-1 molecules are believed to localize to the tip of microvilli in order to facilitate efficient binding of PSGL-1 to selectins[20,33,34]. Previous work from our lab also suggested the localization of CD44 to the microvilli of the cells[22]. Our findings in this study, in principle, agree with this model that the spatial confinement of the selectin ligands to microvilli enhances the probability that multiple ligand molecules in the confined area bind to P- or E-selectin on the surface, which leads to the formation of a tether against the tension exerted to the tethering point. Simulation studies suggested that spatial clustering of ligands strengthens ligand-receptor interactions and helps cells to roll stably on a selectin surface[35,36]. While these studies did not consider the formation of the tethers and slings, the critical role of the spatial clustering of selectin ligands in the cell rolling reported in the study is consistent with our observation. Our finding also suggests that the microvilli play a key role in the spatial clustering of the selectin ligands and the formation of the tethers and slings during cell rolling over E-selectin.

Data from our lab has shown that the binding of E-selectin to its ligands is limited by a slow-on rate[23,25] and this observation that clustering of the ligands (i.e., CD44 and PSGL-1) occurs at anchor points helps to explain how the cell in flow can overcome this "slow on" rate and enhance the binding of E-selectin to its ligands ultimately through the increase in the local concentration of the ligands.

**Selectin ligands are spatially confined to the tethers and slings**. Based on these findings, we next sought to investigate the mechanism of the formation of the tethers. To that end, we calculated the total number of the PSGL-1 molecules on a single tether during its elongation (Fig. 6a). The time-lapse images of PSGL-1 showed that the integrated fluorescence intensity per unit length (i.e., number density of the PSGL-1 molecules along the tether) decreases (Fig. 6b) during the elongation of the tether (Fig. 6c), whereas the integrated intensity over the entire tether (i.e., total number of the PSGL-1 molecules on the tether) is almost constant during the tether elongated from 75 to 110 μm (Fig. 6d).

We also calculated the mean total number of the PSGL-1 molecules on single tethers and slings at different shear stresses (Fig. 6e) using the mean lengths of the tethers/slings (Fig. 3b, f) and the mean distances between the adjacent PSGL-1 molecules

on the tethers/slings at different shear stresses (Fig. 6g, Supplementary Fig. 27). We found that both the mean lengths and distances increased with the increase of the applied shear stresses (Fig. 6f, g). The calculated mean total number of the PSGL-1 molecules on single tethers and slings clearly showed that the number is almost constant over the 1–8 dyne cm$^{-2}$ (0.1–0.8 Pa) although the tethers and slings were two times longer at 8 dyne cm$^{-2}$ (0.8 Pa) compared with those at 1 dyne cm$^{-2}$ (0.1 Pa) (Fig. 6e). The density of CD44 molecules on the tethers and slings also decreased with an increase in the shear stress (i.e., larger distances between the adjacent CD44 molecules on the tethers/slings at higher shear stresses, Supplementary Note 3), similar to PSGL-1 on the tethers and slings.

Given the localization of PSGL-1 and CD44 on the microvilli[20,22,33,34], our results strongly suggest that single tethers and therefore slings are formed from single microvilli upon their binding to the surface E-selectin. This is supported by our observation that the tethers and slings are always covered entirely by CD44 and PSGL-1 (Figs. 2a and 4b). If the tethers are formed from multiple microvilli, this would result in discrete patches of CD44 and PSGL-1 clusters on the tethers and slings.

Previous studies suggested that the localization of selectin ligands to microvilli enables them to interact with selectins during the initial step of homing[33,34]. Our findings here extend this to the tethers and slings (Fig. 6h). During cell rolling, CD44 and PSGL-1 are spatially redistributed to the entire tethers and slings by the conversion of single microvilli to the tethers. This redistribution greatly increases the exposure of these selectin ligands to the surface E-selectin, promoting clustering and efficient selectin-ligand binding that leads to mechanisms that help overcome the slow-on rate leading to slow and stable rolling[23,25].

**Fast diffusion of selectin ligands on tethers and slings promotes stable cell rolling**. Single-molecule imaging revealed that the PSGL-1 molecules show diffusional motion on the tethers and slings (Fig. 7a, Supplementary Fig. 28, Supplementary Note 7, Supplementary Movies 4 and 5). We analyzed their motion using mean square displacement (MSD) analysis of the single-molecule diffusion trajectories (Fig. 7b, Supplementary Figs. 29, and Supplementary Note 8, Supplementary Fig. 30). The MSD analysis showed that the PSGL-1 molecules on the tethers have a mean diffusion coefficient of ~0.17 μm$^2$ s$^{-1}$ (Fig. 7c, Supplementary Fig. 31). The observed diffusion coefficients of the PSGL-1 molecules on the tethers and slings are much faster than those reported for membrane proteins including PSGL-1 on neutrophils (0.003 μm$^2$ s$^{-1}$)[37]. The MSD analysis also demonstrated that the PSGL-1 molecules display random diffusion on the tethers and slings (i.e., linear relationship between the MSD and time lag, Fig. 7b, Supplementary Note 9).

We also conducted single-molecule tracking experiments of the PSGL-1 molecules on the control cells (i.e., localized on microvilli) (Fig. 7d). The experiment was conducted by placing the immunostained cells on the surface of the microfluidic chamber. The MSD analysis of the diffusion trajectories showed that the PSGL-1 molecules on the control cells diffuse much slower than those distributing on the tethers and slings (i.e., mean diffusion coefficient of 0.094 μm$^2$ s$^{-1}$, Fig. 7e). The analysis also revealed that the PSGL-1 molecules on the control cells show a confined diffusion with a confinement area of 0.29 μm$^2$ (Fig. 7b, f, Supplementary Figs. 32, 33, Supplementary Note 10). Due to the distinct rates and modes of the diffusion, the PSGL-1 molecules on the tethers and slings cover much larger displacement areas compared with those restricted to the microvilli (0.14 and 0.26 μm$^2$ for tethers and slings, respectively compared with 0.038 μm$^2$

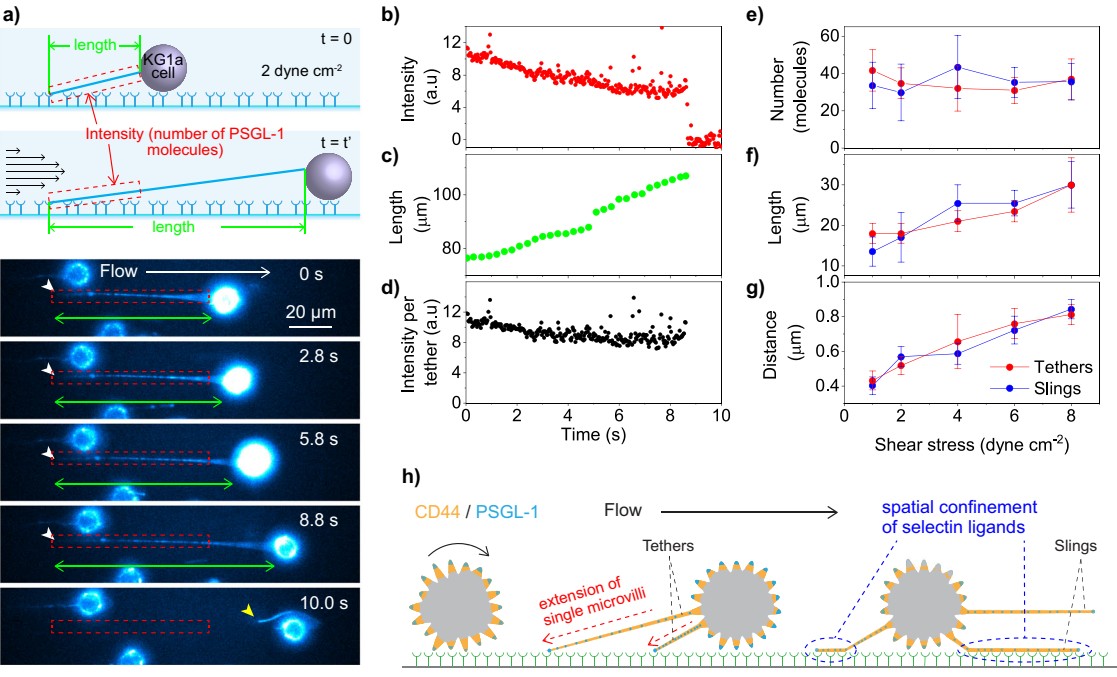

**Fig. 6 Spatial confinement of the selectin ligands to the tethers and slings. a** Time-lapse fluorescence images of the PSGL-1 molecules (immunostained by Alexa-Fluor-555-conjugated anti-PSGL-1 antibody) on the KG1a cell captured during cell rolling over E-selectin at a shear stress of 2 dyne cm$^{-2}$ (0.2 Pa). The white arrowheads show the tethering point. The yellow arrowhead shows the tether detached from the surface E-selectin. The green arrows show the lengths of the tethers while the red dashed boxes show the areas from which the integrated intensities were calculated. The top illustration schematically shows the analysis. **b** Time-lapse integrated fluorescence intensities of the PSGL-1 molecules obtained from the region designated by the red dashed boxes in (**a**). **c** Time-dependent change in the length of the tether obtained from the time-lapse fluorescence images of PSGL-1 displayed in (**a**) green arrows. **d** Time-dependent change in the integrated fluorescence intensity of the PSGL-1 molecules obtained from the whole tether formed during cell rolling displayed in (**a**). **e** Shear stress dependence of the mean numbers of the PSGL-1 molecules per tether and sling. **f** Shear stress dependence of the mean lengths of the tether and sling. The error bars show the standard deviations determined by 19–58 tethers or slings. **g** Shear stress dependence of the mean distances between adjacent PSGL-1 molecules on the tether and sling. The error bars show the standard deviations determined by 23–66 inter-spot distances. **h** Schematic illustration describing how the spatial confinement of the selectin ligands into the tethers and slings due to the formation of individual tethers from single microvilli promote stable and slow rolling of the cell on E-selectin.

for microvilli) during the several hundreds of milliseconds of the lifetime of the tethers[38,39]. The difference becomes even larger when more stable structures (i.e., tethers with multiple anchoring points and non-retracting slings that exist for more than several seconds) are formed during cell rolling.

We found that multiple tethers sometimes merged into a single tether during cell rolling (Fig. 8a, Supplementary Movie 7), which occurs within a second. Together with the larger motional freedom (i.e., faster and random diffusion) of the PSGL-1 molecule on the tethers and slings compared with those localized to the microvilli, these observations imply the involvement of the actin cytoskeleton in the spatiotemporal dynamics of the tethers and slings. We thus conducted a two-color fluorescence imaging experiment of CD44 (labeled by either Alxa-Fluor-488- or Alxa-Fluor-647-conjugated antibodies) and actin (labeled by either silicon-rhodamine-conjugated jasplakinolide[40] or Alexa-Fluor-488-conjugated phalloidin). The fluorescence image of CD44 and actin on the control KG1a cells showed perfect colocalization (Fig. 8b). On the other hand, we did not observe any fluorescence signal of actin from the tethers and slings in most cells rolling over E-selectin. In some rare cases, we found that actin exists along the tethers and slings of the rolling cells, but in a form of small fragmented patches (Fig. 8c). This result unambiguously demonstrates that the cell membrane is detached from the cortical actin cytoskeleton during the formation of the tethers, consistent with previous force spectroscopy studies that indicated the detachment[41–43]. Since PSGL-1 and CD44 on the control cells

are directly or indirectly anchored to the actin cytoskeleton in the microvilli by actin-binding ERM proteins[32,34,44], the formation of the tethers and therefore the detachment of the membrane from the actin cytoskeleton would cause a significant enhancement in the motional freedom of PSGL-1 and CD44, consistent with the results obtained from the single-molecule tracking analysis (Fig. 7b). Previous work from our lab using micorfluidics-based super-resolution fluorescence microscopy clearly shows that rolling causes significant reorganization of the clustering behavior of CD44 and PSGL-1, from patchy to elongated network-like structures all over the surface of the cell[22]. Moreover, in these elongated structures CD44 and PSGL-1 colocalized with actin which supports previous studies implicating the importance of PSGL-1 binding to actin for stable rolling[32]. Despite the numerous elongated structures we saw[22], here we observed that only a few of these lead to tethers and slings and thus detach from the actin cytoskeleton to allow the motional freedom of the selectin ligands. It is possible that the restructuring of the microvilli into these elongated actin-filled structures is a prerequisite to the formation of tethers and slings that detach from actin. Thus, we suspect that both the elongated structures and those that lead to tethers and slings contribute to the slow and stable rolling of cells.

To form a selectin-ligand bond, the ligand molecules on the rolling cell have to be located within tens of nanometers distance from the surface E-selectin molecules[45]. If the locations of the PSGL-1 molecules are fixed, the probability to find the surface

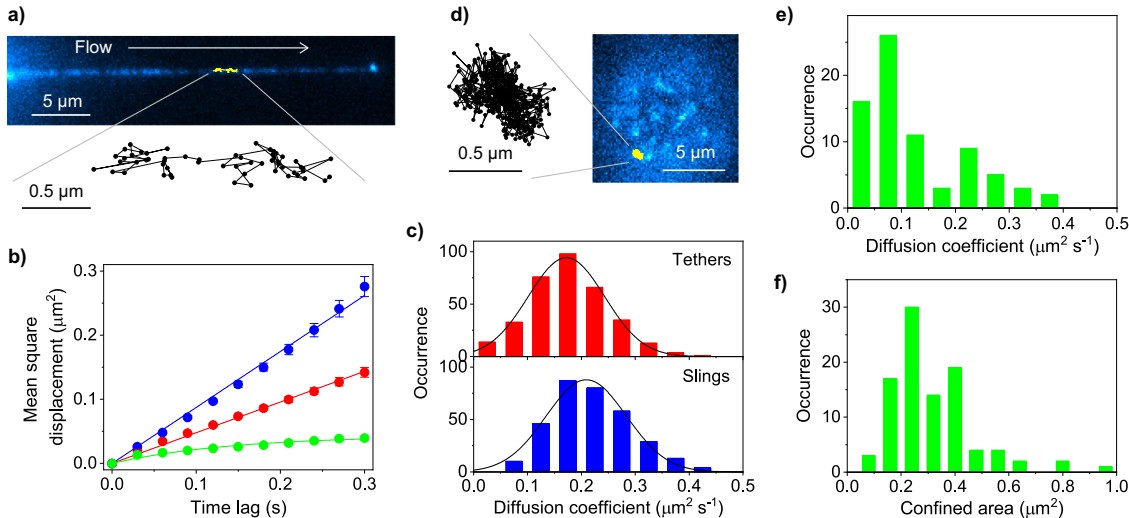

**Fig. 7 Single-molecule tracking analysis of PSGL-1 on the tethers and slings. a** Single-molecule fluorescence images of PSGL-1 (immunostained by Alexa-Fluor-555-conjugated anti-PSGL-1 antibody) on the sling of the KG1a cell formed during cell rolling over E-selectin at a shear stress of 2 dyne cm$^{-2}$ (0.2 Pa). An example of the single-molecule diffusion trajectories obtained from the time-lapse fluorescence images is shown by the yellow line. **b** Mean square displacement (MSD) versus time lag plots obtained for the PSGL-1 molecules diffusing on the tethers (red), slings (blue), and localized on the microvilli of the control KG1a cells (green). The error bars show the standard errors of the mean determined by 340, 320, and 129 MSD plots obtained for the PSGL-1 molecules diffusing on the tethers, slings, and localized on the microvilli of the control KG1a cells. **c** Frequency histograms of the diffusion coefficient of the PSGL-1 molecules on the tethers (top) and slings (bottom) of the KG1a cells formed during cell rolling over E-selectin at a shear stress of 2 dyne cm$^{-2}$ (0.2 Pa). The diffusion coefficients were calculated by fitting the MSD plots obtained from the individual diffusion trajectories to Eq. 4. The solid lines show Gaussian fittings. **d** Single-molecule fluorescence images of PSGL-1 (immunostained by Alexa-Fluor-488-conjugated anti-PSGL-1 antibody) on the microvilli of the control KG1a cell. An example of the single-molecule diffusion trajectories obtained from the time-lapse fluorescence images is shown in the yellow line. **e** Frequency histogram of the diffusion coefficient of the PSGL-1 molecules on the microvilli of the control KG1a cells. The diffusion coefficients were calculated by fitting the MSD plots obtained from the individual diffusion trajectories to Eq. 5. **f** Frequency histogram of the confined area of the diffusing PSGL-1 molecules on the microvilli of the control KG1a cells. The confined areas were calculated by fitting the MSD plots obtained from the individual diffusion trajectories to Eq. 5. All the fluorescence images of the rolling cells were captured by injecting KG1a cells into the rh E-selectin-deposited microfluidic chambers at a shear stress of 2 dyne cm$^{-2}$ (0.2 Pa).

E-selectin molecules within this distance is relatively low. Thus, the fast and random motion of the selectin ligands on the tethers and slings, which enables them to scan large surface area during a limited lifetime of tethers (with anchoring points) and slings, should facilitate efficient binding of the selectin ligands to the surface E-selectin and thus enable slow and stable rolling of the cell (Fig. 9).

In conclusion, our results suggest that during the initial step of HSPC and leukemic cell migration, the selectin ligands are spatially confined to the tethers and slings through the development of the tethers from single microvilli and their motional freedom on the tethers and slings is enhanced by the detachment of the membrane from actin cytoskeleton during the formation of the tethers. The tether and sling dynamics including the elongation, anchoring, long lengths and multiplicity of these structures altogether are used by the cells to achieve slow rolling at low to high shear stresses. These mechanisms enable the efficient utilization of the limited amount of the selectin ligands, which contributes to overcome the slow on rate of binding to the underlying E-selectin[23,25] to achieve effective selectin-ligand binding, resulting in this slow and stable cell rolling. Our results demonstrate that the microfluidics-based single-molecule imaging provides a unique experimental platform to investigate cellular interactions at the molecular level under the presence of external force.

## Methods

**Cell culturing and treatments**. Human acute myelogenous leukemia cell lines, KG1a cells (ATCC CCL-246.1) were maintained in the lab using cell culturing facility in RPMI 1640 (1×) media (Gibco) that contains 10% fetal bovine serum

(Corning) and antimicrobial agents (100 U ml$^{-1}$ penicillin, 100 µg ml$^{-1}$ streptomycin, Hyclone) incubated at 37 °C in presence of controlled 5% CO$_2$. For the disruption of lipid raft microdomains, the KG1a cells were treated with MβCD. Briefly, $10^6$ ml$^{-1}$ KG1a cells were washed three times by HBSS buffer (pH = 7.0–7.4) and incubated in serum-free Iscove's modified Dulbecco's medium (IMDM, Gibco) with 10 mM HEPES buffer (Sigma) and 10 mM MβCD (Sigma) at 37 °C for 30 min. The viability of the cells after the treatment was confirmed by trypan blue staining. For siRNA knockdown of CD34 in KG1a cells, KG1a cells were first collected and treated for 20 min at 37 °C with bromelain (250 µg ml$^{-1}$; Sigma) in RPMI media with 10% FBS. Cells were then washed twice with PBS and prepared for transfection. The bromelain treated KG1a cells were then transfected using 250 pmol of CD34 siRNA (Silencer Select, Life technologies; 4392420-s2644) or 250 pmol of a scrambled control siRNA (Silencer Select, Life technologies; 4390843). Transfection was done using the SE Cell Line 4D-Nucleofector$^{TM}$ X Kit (Lonza). The company protocol was followed to achieve the knockdown. Cells were then collected after 48 h and prepared for Western blot analysis to determine the extent of the CD34 inhibition.

**Preparation of human CD34$^{pos}$-HSPCs**. Primary human CD34$^{pos}$-HSPCs isolated from umbilical cord blood (CD34$^{pos}$-HSPCs) and the mononuclear cells (MNC) from whole cord blood were purchased from ALL Cells (USA). For CD34$^{pos}$-HSPCs isolation from whole cord blood, the MNCs were washed and filtered through a 100 µm cell strainer (BD Falcon) followed by lineage depleton by negative selection using a lineage cell depletion cocktail (Milteny Biotec). The lineage depleted fraction was stained with 13 markers for lineage-specfic monoclonal antibodies FITC (CD2, CD3, CD4, CD11b, CD14, CD15, CD24, CD10, CD41, CD20, CD66c, CD127, CD33) (Biolegend and BD) and Pacific blue conjugated anti-CD34 (Biolegend), for 30 min at 4 °C. Before sorting, 7-AAD was added to the tubes. Afterwards, 13Lin−CD34$^{pos}$ cells were collected by the fluorescence-activated cell sorter (FACSAria, BD Biosciences, San Jose, CA, USA).

For CD34$^{pos}$-HSPCs, the cells were thawed using stem span SFEM II medium (Stem Cell Technologies, Canada) and washed 1× with the same medium. The cells were resuspended in 1 mL of medium contained 1× Stem Span Cytokine Cocktail CC100 (Stem Cell Technologies, Canada) in 24-well plate, then the cells were analyzed within 24 h.

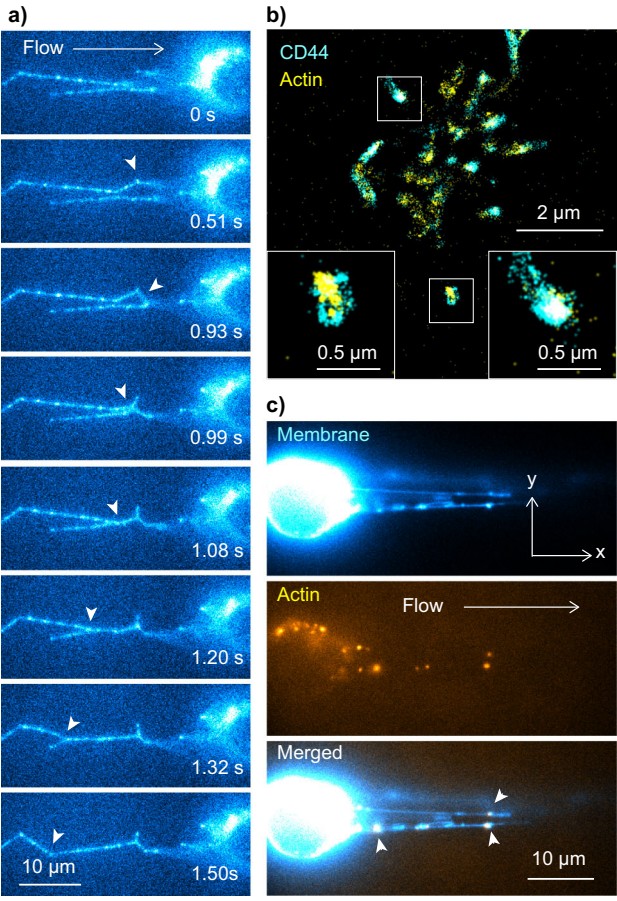

**Fig. 8 Detachment of cell membrane from actin cytoskeleton during the tether formation. a** Time-lapse fluorescence images of the PSGL-1 molecule (immunostained by Alexa-Fluor-555-conjugated anti-PSGL-1 antibody) on the KG1a cell captured during cell rolling over E-selectin, which show the merger of the multiple tethers. The white arrowheads show the regions in which the merger of the tethers occurs. **b** Two-color super-resolution fluorescence localization microscopy image of CD44 (cyan, immunostained by anti-CD44 515 antibody followed by Alexa-Fluor-647-conjugated secondary antibodies) and actin (yellow, labeled by Alexa-Fluor-488-conjugated phalloidin) on the control KG1a cell. The insets show enlarged views of the yellow regions. **c** Two-color fluorescence images of the cell membrane (cyan, stained by Vybrant DiO dye) and actin (yellow, labeled by silicon-rhodamine-conjugated jasplakinolide) on the slings of the KG1a cells formed during cell rolling over E-selectin. The white arrowheads show the regions in which the patches of actin colocalize with the slings. All the fluorescence images of the rolling cells were captured by injecting KG1a cells into the microfluidic chambers at a shear stress of 2 dyne cm$^{-2}$ (0.2 Pa) whose surface was coated by the rh E-selectin at a density of 15 molecules μm$^{-2}$.

**Preparation and functionalization of the microfluidic chambers.** Commercially available microfluidic chambers, μ-slide VI 0.1 uncoated (1 mm width and 100 μm height, ibidi GmbH) were used for in vitro cell rolling assay. We calculated applied shear force ($\tau$) using the following equation, $\tau = 6\mu Q/a^2 b$, where $a$ and $b$ are the height and width of the chamber and $\mu$ and $Q$ are the viscosity of the medium and volumetric flow rate. Recombinant homodimeric human (rh) E-selectins (Sino Biological) were deposited on the inner surface of the microfluidic chambers by incubating with 20 μl of the rh E-selectin (0.5–4 μg ml$^{-1}$) at 4 °C for overnight. After the incubation, the chambers were washed three times using 1× Hank's balanced Salt Solution (HBSS, Gibco) and blocked with 1% bovine serum albumin (BSA, Sigma Aldrich) by incubating the chambers at 4 °C for 1-2 h. The chambers were then washed by 1× HBSS three times.

**Estimation of the surface density of the rh E-selectin.** A monoclonal anti-human E-selectin antibody (HCD62E clone, Biolegend) was labeled with Alexa-

Fluor-647 dye as described below. The Alexa-Fluor-647-conjugated anti-E-selectin was diluted into HBSS at the concentration of 10 μg ml$^{-1}$. Twenty microliters of the Alexa-Fluor-647-conjugated anti-E-Selectin antibody was introduced into the microfluidic chambers on which the homodimeric rh E-Selectin was deposited. The chambers were incubated with the antibody at 4 °C for 1 h. The fluorescence from the Alexa-Fluor-647-conjugated anti- E-Selectin antibody bound to the surface rh E-selectin was recorded using a custom-built single-molecule fluorescence imaging setup (see below). The surface density of the rh E-selectin was calculated by comparing the integrated fluorescence intensity obtained from a unit area (1 μm$^2$) of the rh E-selectin-deposited surface with the intensity obtained from single rh E-Selectin molecules labeled by the Alexa-Fluor-647-conjugated anti-E-Selectin antibody. The fluorescence intensity from single rh E-Selectin molecules labeled by the antibodies was determined by depositing the Alexa-Fluor-647-conjugated anti-E-Selectin antibody on the rh E-selectin-deposited surface at a very low concentration (0.02 μg ml$^{-1}$) and measuring the fluorescence intensities of the individual fluorescent spots obtained from the single rh E-selectin molecules labeled by the antibodies.

**Cell rolling assay using bright-field microscope.** The rh E-selectin-deposited microfluidic chambers were connected to a syringe pump (Harvard Apparatus, PHD Ultra) using a silicone tubing (inner diameter of 0.8 mm, ibidi GmbH) and mounted on an inverted optical microscope (Olympus, CKX41). KG1a cells were suspended in HBSS buffer containing 0.7 mM of Ca$^{2+}$ (anhydrous CaCl$_2$, Sigma Aldrich) and 1% (w/v) BSA. The rolling behavior of the cells in this perfusion buffer was observed using the inverted light microscope that is equipped with a 20× objective lens (Olympus, LCAch N 20X PHP) and XC10 CCD camera (Olympus). The transmitted images were recorded by the CCD camera using CellSens imaging software provided by Olympus. All images were recorded at the frame rate of 15 fps with 30 ms exposure time. The pixel size of the CCD camera is 6.45 μm. The KG1a cells were perfused into the microfluidic chamber mounted on the microscope stage at the flow rate of 100 dyne cm$^{-2}$ (10 Pa), then the flow rate was decreased gradually to reach a constant shear stress of 4 dyne cm$^{-2}$ (0.4 Pa), 2 dyne cm$^{-2}$ (0.2 Pa), 1 dyne cm$^{-2}$ (0.1 Pa), 0.5 dyne cm$^{-2}$ (0.05 Pa), or 0.25 dyne cm$^{-2}$ (0.025 Pa) where cell rolling behavior was monitored for 78 s in each experiment. The rolling velocities were calculated using CellSens or TrackMate, an Image J plugin. The rolling velocities were estimated by measuring the distance traveled by each cell after being bound to the immobilized rh E-selectins. The number of the bound cells was determined by counting the number of the attached cells in five or six field of views (0.444 mm × 0.335 mm size) in at least two independent experiments.

**Fluorescence labeling of the antibodies.** The fluorescence labeling of the antibodies by Alexa-Fluor-488 dye (Invitrogen), Alexa-Fluor-555 dye (Invitrogen), and Alexa-Fluor-647 dye (Invitrogen) was conducted by following the manufacturer's instructions. The labeled antibodies were mouse anti-human E-selectin (HCD62E clone), mouse anti-human CD44 (515 clone, Ms IgG1, κ, BD Bioscience) and mouse anti-human PSGL-1 (KPL-1 clone, Ms IgG1, κ, BD Bioscience) and mouse anti-human CD34 (QBEND 10 clone, Ms IgG1, Bio-Rad). Briefly, the Alexa-Fluor dyes conjugated to N-hydroxysuccinimide (NHS) were dissolved in dimethylsulf-oxide (DMSO, Sigma-Aldrich, 276855) to a final concentration of 10 mg ml$^{-1}$. The antibodies were dissolved in 0.1 M sodium bicarbonate (pH = 8.3–8.4) at a concentration of 6.5 mg ml$^{-1}$. 30.7 μl of the antibody solution and 2 μl Alexa-Fluor dye solution (i.e., mixing ratio of ten to one, which corresponds to the dye to antibody molar ratio of ~120–1) were mixed and incubated for 1 h at room temperature. After the labeling reaction, 467 μl of HBSS was added to the reaction mixture. Free dyes in the mixture were removed by using a spin filter (Pall Corporation, Nanosep centrifugal devices with Omega membrane, OD010C34) at a centrifuge speed of 10,000 × $g$ for 5 min. The solution was resuspended in HBSS and the centrifugation was repeated several times. The final concentration of the antibodies and the labeling degree were determined by UV–vis absorption spectra of the Alexa-Fluor dye-conjugated antibodies. The degree of labeling was in the range of 4–9 dyes per antibody (Supplementary Figs. 34 and 35, Supplementary Note 11).

**Fragmentation and fluorescence labeling of the antibodies.** For the digestion and purification of the anti-PSGL-1 antibody (KPL-1 clone, IgG1), we used Pierce Mouse IgG1 Fab and F(ab')2 Preparation Kit or Pierce Mouse IgG1 Fab and F(ab')2 Micro Preparation Kit (Thermo Scientific). The kit uses immobilized ficin protease to efficiently digest mouse IgG1 into Fab or F(ab')2 fragments, depending on the concentration of cysteine and solution pH. Briefly, the digestion buffer was prepared by dissolving 43.9 mg of cysteine•HCl in 10 ml of the supplied Mouse IgG1 digestion buffer (pH = 5.6) to produce Fab fragments. Then, the immobilized enzyme was dispensed into the spin column, centrifuged, and washed using the prepared digestion buffer. The antibody (1 mg ml$^{-1}$, 125 μl) was desalted using the accompanied Zeba column. The flow through of at least 100 μl that contained the antibody was incubated with the immobilized enzyme for 3–5 h at 37 °C on an end-over-end mixer, which digested the antibody into Fab fragments. After the incubation, the column was put into a 2 ml collection tube and centrifuged at 5000 × $g$ for 1 min to collect the digested antibodies. The collected flow through was incubated with protein A column for 10 min at room temperature on an end-over-end mixer, followed by centrifugation at 1000 × $g$ for 1 min to separate the Fab

**Fig. 9 Mechanistic model of leukemic cell rolling.** Schematic illustration describing how the faster and random diffusional motion of the selectin ligands on the tethers and slings, due to the detachment of the cell membrane from the cortical actin cytoskeleton during the tether formation, promote stable and slow rolling the cell on E-selectin.

fragments from Fc and undigested antibodies. The flow through that contained Fab fragments was collected. After the fragmentation and purification, concentrations of the anti-PSGL-1 antibody (KPL-1 clone, IgG1) and its fragments were determined by measuring the absorbance at 280 nm using UV–vis spectrophotometer (Thermo Scientific, NanoDrop 2000).

Fluorescence labeling of the Fab fragment of the anti-PSGL-1 antibody by the Alexa-Fluor dyes conjugated to N-hydroxysuccinimide (NHS) was conducted in a manner similar to the labeling of the whole antibody. The Fab fragment and the dye were mixed at the mixing ratio of thirty to one, which corresponds to the dye to antibody molar ratio of ~120–1. The degree of labeling with this mixing ratio was in the range of 1–3 dyes per Fab fragment.

**Fluorescence labeling of the primary cells and KG1a cells**. Prior to immunostaining of the KG1a and primary cells, the cells were incubated with Fc blocker (0.15–0.3 ml for each $1 \times 10^6$ cell, Innovex biosciences) on ice for 45 min, then washed three times with 1× HBSS. The cells were immunostained by either Alexa-Fluor-488-conjugated anti-PSGL-1, Alexa-Fluor-555-conjugated anti-PSGL-1, Alexa-Fluor-647-conjugated anti-PSGL-1, Alexa-Fluor-488-conjugated anti-CD44, or Alexa-Fluor-647-conjugated anti-CD44 for 45 min on ice at a concentration of 5 μg for each $10^6$ cells. The immunostained cells were suspended in a Fluorobrite (Gibco) perfusion buffer containing 0.7 mM of $Ca^{2+}$ at the density of $0.5–1 \times 10^6$ cells ml$^{-1}$. Then, the cells were injected into the rh E-selectin-deposited microfluidic chambers using the silicone tubing connected to the syringe pump. The KG1a or primary cells were perfused into the microfluidic chamber mounted on the stage of an inverted fluorescence microscope (Olympus, IX71) at the flow rate of 100 dyne cm$^{-2}$ (10 Pa). Then the flow rate was decreased gradually to reach a constant shear stress of 1–8 dyne cm$^{-2}$ (0.1–0.8 Pa) where real-time fluorescence images of CD44 or PSGL-1 on the live KG1a or primary cells were recorded by using EMCCD camera connected to the iQ3 software (see below).

Two-color fluorescence imaging of CD44 and PSGL-1 on the KG1a cells were conducted by immunostaining both CD44 and PSGL-1 with a combination of either Alexa-Fluor-488-conjugated anti-CD44 and Alexa-Fluor-647-conjugated anti PSGL-1 or Alexa-Fluor-647-conjugated anti-CD44 and Alexa-Fluor-488-conjugated anti PSGL-1. For the two-color immunostaining, the KG1a cells were incubated simultaneously with the two antibodies (5 μg of each antibody for $10^6$ cells) on ice for 45 min.

Two-color fluorescence imaging of CD44 and CD34 on KG1a and CD34 knockdown cells were conducted by immunostaining both CD44 and CD34 with a combination of Alexa-Fluor-647-conjugated anti CD44 and Alexa-Fluor-488-conjugated anti-CD34. For the two-color immunostaining, KG1a cells were first incubated with CD44 antibody on ice for 45 min and then with CD34 antibody (5 μg of each antibody for $10^6$ cells) on ice for 45 min.

We used Vybrant DiO (Molecular Probe) for staining the plasma membrane of KG1a cells. The harvested and counted KG1a cells were washed three times with serum-free 1× IMDM. The staining solution was prepared by adding 1 μl of the Vybrant DiO stain to each 1 ml serum-free IMDM. The $10^6$ cells after washing were suspended in the 1 ml staining solution, mixed and incubated at 37 ºC for 15 min. The stained cells were washed twice in 1× serum free IMDM and once with 1× HBSS and resuspend in the Fluorobrite perfusion buffer containing 0.7 mM of $Ca^{2+}$ at the density of $0.5–1 \times 10^6$ cells ml$^{-1}$. For the two-color fluorescence imaging of the cell membrane and the selectin ligands (either CD44 or PSGL-1), the Vybrant DiO-stained KG1a cells were incubated with the Alexa-Fluor-647-conjugated anti-CD44 or anti-PSGL-1 antibodies (5 μg for each $10^6$ cells) on ice for 45 min.

We used silicon rhodamine (SiR)-conjugated jasplakinolide (Cytoskeleton, Inc.) for staining the actin cytoskeleton of the live KG1a cells. The harvested KG1a cells were washed three times with 1× HBSS. The staining solution was prepared by adding 1 μl of DMSO containing1 mM of SiR-jasplakinolide and 1 μl of DMSO containing 10 mM of verapamil into 998 μl Dulbecco's modified Eagle medium (DMEM) containing 10% bovine serum. The $10^6$ cells after washing were suspended in the 1 ml staining solution, mixed, and incubated at 37 ºC for 1 h in presence of 5% $CO_2$. The stained cells were washed three times using 1× HBSS and resuspended in the Fluorobrite perfusion buffer containing 0.7 mM of $Ca^{2+}$ at the density of $0.5–1 \times 10^6$ cells ml$^{-1}$. For the two-color fluorescence imaging of the

actin cytoskeleton and cell membrane, the SiR-stained (i.e., actin-labeled) KG1a cells were incubated with Vybrant DiO stain at 37 ºC for 15 min.

The fluorescence imaging experiment of CD44 and cell membrane on the MβCD-treated live KG1a cells was conducted in a way similar to that on the control KG1a cells by using Vybrant DiO for the membrane stain and Alexa-Fluor-647-conjugated anti-CD44 for the immunostaining of CD44. The fluorescence labeling was conducted immediately after treating the cells with MβCD.

We used Alexa-Fluor-488-conjugated phalloidin (molecular Probes-Thermo Fisher Scientific) and anti-CD44 for staining actin cytoskeleton and CD44 of the fixed KG1a cells. The harvested KG1a cells were fixed in 3% (w/v) paraformaldehyde (Electron Microscopy Sciences) and 0.2% (w/v) glutaraldehyde (Electron Microscopy Sciences) in HBSS for 20 min at room temperature. The fixed cells were blocked with 10% goat serum (Sigma) for 40 min at 37 ºC. The cells were then incubated with the anti-CD44 antibody (15 μg ml$^{-1}$) diluted in 2% BSA in HBSS for 40 min at 37ºC, followed by the Alexa-Fluor-647-conjugated goat anti-mouse secondary antibody (5 μg ml$^{-1}$) diluted in 2% BSA in HBSS for 40 min at 37ºC. The immunostained cells were fixed again in 3% (w/v) paraformaldehyde and 0.2% (w/v) glutaraldehyde in HBSS for 10 min at room temperature. The cells were then permeabilized in 0.1% Triton X-100 (Sigma-Aldrich) in cytoskeleton buffer (10 mM MES (pH = 6.1), 150 mM NaCl, 5 mM EGTA, 5 mM glucose, and 5 mM MgCl$_2$) for 10 min at room temperature. The permeabilized cells were labeled for actin cytoskeleton with freshly prepared 0.5 μM Alexa-Fluor-488-conjugated phalloidin in the cytoskeleton buffer containing 1% BSA for 1 h at room temperature. Then, the cells were fixed again in 3% (w/v) paraformaldehyde and 0.2% (w/v) glutaraldehyde in HBSS for 10 min at room temperature. The cells were injected into poly-L-ornithine (Sigma)-coated microfluidic chambers (ibidi GmbH, sticky-slide VI 0.4) using the syringe pump and incubated overnight at 4 ºC.

The fluorescence imaging experiment of CD44 on the MβCD-treated fixed KG1a cells was conducted in a way similar to that on the fixed control KG1a cells by using a combination of the anti-CD44 antibody and the Alexa-Fluor-647-conjugated anti-mouse secondary antibody[22]. The cells were injected into poly-L-ornithine-coated microfluidic chambers (ibidi GmbH, sticky-slide VI 0.4) using the syringe pump and incubated overnight at 4 ºC.

For single-molecule tracking experiment of the PSGL-1 molecules on the control live KG1a cells (i.e., not rolling cells), the cells were blocked with Fc receptor blocker (Innovex Biosciences) for 45 min on ice, then the cells were immunostained by Alexa-Fluor-488-conjugated anti-PSGL-1 for 45 min on ice at much lower concentration (20 ng for each $10^6$ cells) to sparsely label the PSGL-1 molecules localized on microvilli so that the motion of individual PSGL-1 molecules can be captured. The immunostained KG1a cells were suspended in HBSS at the density of $0.5–1 \times 10^6$ cells ml$^{-1}$. Then, the cells were injected into the microfluidic chambers (ibidi GmbH, glass-slide VI 0.5).

**Fluorescence imaging**. Fluorescence imaging experiments were conducted using a home-built wide-field fluorescence microscopy setup[46]. Continuous-wave (CW) solid-state laser operating at either 488 nm (60 mW, Cobolt, MLD), 532 nm (100 mW, Cobolt, Samba), or 638 nm (60 mW, Cobolt, MLD) that passed an excitation filter (Semrock, LD01-640/8, FF01-532-12.5 or FF01-488/6 for the 638 nm, 532 nm or 488 nm excitation, respectively) and a beam expander (Thorlabs) was introduced into an inverted microscope (Olympus, IX71) through a focusing lane ($f = 300$ mm). The laser was reflected by a dichroic mirror (Semrock, FF660-Di02-25×36, Di01-R532-25 × 36, or FF506-Di03-25 × 36 for the 638, 532 or 488 nm excitation, respectively) and the sample was illuminated through an objective lens (Olympus, 100× NA = 1.49, UAPON 100XOTIRF, 60× NA = 1.3, UPLSAPO60XS2 or 40× NA = 1.25, UPLSAPO40XS). The excitation power was adjusted to 1–2 mW cm$^{-2}$ at the sample plane using an acousto-optical tunable filter (AOTF; AA Optoelectronics) inserted in the excitation beam pass. The fluorescence from the sample was captured by the same objective, separated from the illumination light by the same dichroic mirror, passed an emission bandpass filter (Semrock, FF01-697/58-25, FF01-580/60, or FF01-550/88 for the 638, 532 or 488 nm excitation, respectively), and detected by an EMCCD camera (Andor Technology, iXon3 897). The fluorescence images were recorded with either 133-nm or 333-nm pixel size at 30 ms exposure time. The exposure of the EMCCD camera and the illumination of the sample by the excitation laser were synchronized by the AOTF using a laser control system (Andor Technology, PCUB-110).

The image acquisition was done using the Andor iQ3 software. 3D fluorescence images were obtained by recording epi-fluorescence images of the cells at different Z-axis positions with 0.5–1.0 μm step size using a piezo objective scanner (PI PIFPC® P721) and reconstructing 3D images using ImageJ plugin.

Two-color fluorescence imaging experiments were conducted by introducing 638-nm and 488-nm lines of the lasers coaxially into the inverted microscope in the same way as the single-color excitation. The samples were excited through the objective lens (Olympus, 100× NA = 1.49, UAPON 100XOTIRF or 40× NA = 1.25, UPLSAPO40XS). The fluorescence from the sample was captured by the same objective, separated from the illumination light by a multiband dichroic mirror (Semrock, Di03-405/488/561/635-t1-25 × 36), and passed through a TuCam dual-camera adaptor (Andor Technology) equipped with a filter cassette containing a dichroic mirror (Semrock, FF580-FDi01-25 × 36) to separate the fluorescence into two channels. The separated fluorescence from the samples was detected by two EMCCD cameras (Andor Technology, iXon3 897) through emission bandpass filters (Semrock, FF01-550/88-25 and FF01-697/58-25).

**Determination of the number of PSGL-1 molecules on tethers and slings**. The number of PSGL-1 molecules in each spot on the tethers and slings formed during KG1a cells rolling over E-selectin was calculated by comparing the integrated fluorescence intensity obtained from each PSGL-1 spot with the intensity obtained from a single Alexa-Fluor-dye-conjugated anti-PSGL-1 antibody. To determine the fluorescence intensity obtained from a single Alexa-Fluor-dye-conjugated anti-PSGL-1 antibody, the antibody was deposited on the surface of the microfluidic chamber at a concentration of 0.02 μg ml⁻¹. Fluorescence signals from the single deposited antibodies were captured at the experimental conditions identical to those for the imaging experiment on the immunostained KG1a cells rolling over E-selectin. We used only well-focused fluorescence images of the PSGL-1 spots on the tethers and slings for this analysis to ensure an accurate estimation of the number of the PSGL-1 molecules.

**Single-molecule tracking analysis**. Single-molecule tracking analysis of the PSGL-1 molecules moving along the tethers and slings of KG1a cells rolling over E-selectin and localized on the microvilli of control KG1a cells were conducted by generating single-molecule diffusion trajectories using a versatile tracking algorithm[47,48]. The algorithm uses a mixture-model fitting algorithm to localize and track multiple particles in the same field of view. It can detect the merging and splitting of particles during motion. To achieve these tracking targets, the algorithm localizes and tracks all the local maxima in the single-molecule image including maxima that are partially overlapping along the tethers and slings. The diffusion rate (i.e. diffusion coefficient) and mode were analyzed by mean square displacement (MSD) analysis[49]. The MSD was calculated from the generated diffusion trajectories by using Eq. (3).

$$MSD(\triangle t) = \left\langle \left(x_{i+n} - x_i\right)^2 + \left(y_{i+n} - y_i\right)^2 \right\rangle, \quad (3)$$

where $x_{i+n}$ and $y_{i+n}$ demote the spatial positions after time interval $\triangle t$, given by the frame number, $n$, after starting at positions $x_i$ and $y_i$. The MSD-$\triangle t$ profiles were fitted to random (Eq. (4)), and confined (Eq. (5)) diffusion models.

$$MSD(\triangle t) = 4D\triangle t, \quad (4)$$

where $D$ is the diffusion coefficient,

$$MSD = \frac{L^2}{3}\left[1 - \exp\left(\frac{-12D\triangle t}{L^2}\right)\right], \quad (5)$$

where $L$ is the side length of the confined area.

**Super-resolution fluorescence imaging and analysis**. Super-resolution (SR) fluorescence localization microscopy imaging of fixed immunolabeled KG1a cells was performed in an imaging buffer composed of TN buffer (50 mM tris (pH 8.0) and 10 mM NaCl), oxygen scavenging system (glucose oxidase (0.5 mg ml⁻¹, Sigma), catalase (40 μg ml⁻¹, Sigma), 10% (w/v) glucose), and 100 mM β-mercaptoethanol (Sigma) as a reducing reagent. The imaging solution was prepared immediately before the imaging experiments. The imaging experiments were conducted on a custom-built wide-field illumination fluorescence microscope on an inverted optical microscope platform described above[22,50]. The fluorescence images were recorded using a 150 × 150 pixel region of the EMCCD camera with 83-nm pixel size at 10-ms exposure time. The fluorescence image sequences with 10,000 frames were recorded for the reconstruction of SR localization microscopy images.

The SR images were reconstructed by using either a custom-written MATLAB (MathWorks) code or Localizer software[51]. The positions of the AF-647 molecules were determined by 2D Gaussian fitting of the images. We removed fluorescence spots whose width was significantly larger (>200 nm) than the point spread function (PSF) of the optical system (PSF, ~130 nm) from the analysis. The effect of the stage drift in the $xy$ plane was corrected by reconstructing the subimages using 5000 localizations. In the two-color SR imaging, TetraSpeck microspheres (diameter, 100 nm) deposited on a cleaned coverslip was used to calibrate the shift between the two channels. Using fluorescence images of the TetraSpeck microspheres recorded simultaneously on the two cameras, we generated a registration map that corrects the shift between the two images and applied the registration map to the images obtained from the cell samples.

**Scanning electron microscopy**. The harvested KG1a cells (10⁷ cells) were washed twice with HBSS. The cells were fixed in 2-2.5% glutaraldehyde in 0.1 M Cacodylate buffer (pH = 7.2–7.4) in 4 °C for overnight. The fixed cells were washed three times with the 0.1 M Cacodylate buffer with 15 min submerged in buffer before the next wash and resuspended in the 200 μl of the buffer. Then, 100 μl of the cell suspension was added on a cover slip, which was brushed with polylysine and incubated overnight inside a moisturized chamber before use. The cells were further fixed in 1% osmium tetroxide in 0.1 M Cacodylate buffer for one hour in dark. The cells were washed three times with distilled water, where they were submerged in water for 15 min before the next wash, and dehydrated in gradient ethanol (30, 50, 70, 90, and 100%). Samples were carried further onto the Critical Point Drying apparatus and covered with 100% ethanol to completely submerge the samples. The dehydrated cells were placed on a holder of the scanning electron microscope (SEM) and coated with 4 mm platinum (K575X Sputter Coater, Quorum). The SEM images were recorded using Teneo SEM (Thermo Fischer).

**Flow cytometry**. Binding specificity of the mouse anti-human PSGL-1 and mouse anti-human CD44 antibodies, which were used for the fluorescence imaging experiments, to PSGL-1 and CD44 on KG1a cells were evaluated by flow cytometry. KG1a cells (10⁶ cells ml⁻¹) were incubated with either anti-human PSGL-1 antibody (clone KPL-1, Ms IgG1, κ), anti-human CD44 antibody (clone 515 Ms IgG1, κ, or 2C5 Ms IgG2a) or purified mouse IgG1κ isotype (BioLegend) at a final concentration of 10 μg ml⁻¹ in HBSS at 4 °C for 30 min. Subsequently, the KG1a cells were incubated with Alexa-Fluor-488–conjugated goat anti-mouse antibody (5 μg ml⁻¹, IgG, Invitrogen) in HBSS at 4 °C for 20 min. For the secondary antibody control, KG1a cells (10⁶ cells ml⁻¹) were incubated with Alexa-Fluor-488–conjugated goat anti-mouse antibody (5 μg ml⁻¹) in HBSS at 4 °C for 20 min. The fluorescence intensity was determined using a FACSCanto flow cytometer (Beckman Dickenson).

**Western Blot Analysis**. WT KG1a, scrambled, or CD34 siRNA knockdown cells were lysed using a cell lysis buffer containing 88% NP40 (Invitrogen™ Novex™, Fisher Scientific), 10% protease inhibitor (Pierce™, Thermo Scientific), 1% PMSF, and 1% Phosphatase inhibitor (Halt™, Thermo Scientific) at 4 °C for 1 h. The whole-cell lysate was collected and incubated with NuPAGE LDS sample buffer (Invitrogen) and 10% β-mercaptoethanol at 70 °C for 10 min. The samples were then run on an SDS-PAGE gel prior to being transferred to a PVDF membrane. The PVDF membrane was blocked overnight at 4 °C using Tris-buffered saline with Tween-20 (Cell Signaling Technology) containing 5% non-fat skim milk powder. The membrane was then washed and incubated with a mouse anti-human CD34 antibody (QBEND/10, BIO RAD). The membrane was then washed and immunoblotted with HRP-conjugated secondary antibodies prior to imaging.

**Statistics and reproducibility**. Statistical significance was assessed by student's t-test (assuming two-tailed distribution and two-sample unequal variance). All experiments were repeated at least twice to assure reproducibility. All the single-cell fluorescence microscopy images reported in this study are representative examples of multiple ($n > 2$) independent experiments. Further, key behaviors of the cell that were reported in this study (i.e., length of tethers and slings, number of PSGL-1 and CD44 molecules on tethers and slings, diffusion coefficients, and modes of PSGL-1 molecules) were statistically verified.

**Reporting summary**. Further information on research design is available in the Nature Research Reporting Summary linked to this article.

## Data availability

All the source data supporting the findings of this study are available within the paper and in Supplementary Data 1. Any other information is available from the corresponding author upon reasonable request.

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

## Acknowledgements

We thank Ms. Ohoud M. Alharbi for the SEM images of KG1a cells. The research reported in this publication was supported by funding from the King Abdullah University of Science and Technology (KAUST) and the KAUST Office of Sponsored Research under Award No. CRG R2 13 MERZ KAUST 1. We would like to thank Ms. Samar A. Rustum and Ms. Umm Habiba for their support in the management of the lab.

## Author contributions

S.H. and J.S.M. conceived the project. B.A.A. designed the in vitro cell rolling assay and single-molecule fluorescence imaging experiments. K.A.Z. designed the super-resolution microscopy experiments. A.R. prepared cells for SEM imaging? B.A.A. and A.R. conducted the fluorescence imaging experiments with the support of S.N. and analyzed the data with the support of M.F.S. K.A.Z. conducted the super-resolution fluorescence imaging experiments and analyzed the data. B.A.A. conducted the flow cytometry experiments with the support of F.A. M.A. prepared the CD34 knockdown KG1a cells. A.S.A. prepared the primary CD34^pos-HSPCs. M.A., A.S.A., A.R., and S.N. conducted and analyzed the fluorescence imaging experiments related to the CD34 knockdown KG1a cells and the primary human CD34^pos-HSPCs. J.S.M. provided the cell lines. S.H., J.S.M., and B.A.A. wrote the manuscript. All authors discussed the results.

## Competing interests

The authors declare no competing interests.
