## [Peer Review File · Communications Biology]

Reviewers' comments:

Reviewer #1 (Remarks to the Author):

In the manuscript by Bader Al Alwan et al., the authors employ a microfluidic device equipped with ligand-coated chambers to measure the membrane dynamics of rolling cells and, in particular, the diffusion of adhesion receptors (CD44 and PSGL-1). An understanding for the organization and diffusion of receptors that facilitate rolling in the vasculature is indeed incomplete, yet it has long been appreciated that the clustering and the consequential avidity of these interactions is essential. In principle, therefore, this study would pique the interest of a wide-segment of vascular biologists and cell biologists alike. Moreover, single molecule tracking is rarely, if ever done in heterocellular cell-cell contacts or in the context of rolling adhesion biology.

Unfortunately, the authors have opted to use planar, rigid arrays of recombinant E-selectin to coat the plastic chambers attached to their microfluidic devices rather than growing normal endothelial monolayers in which the adhesion ligands/receptors can form meta-stable clusters and interact with the submembrane cytoskeleton. While the authors can mimic an appropriate absolute concentration of ligand in their experiments, they cannot control the orientation, diffusion/fluidity, nor the rigidity of the E-selectin. Therefore, it remains unclear if the phenomenon described inclusive of the tethers and slings would also be observed for the KG1a cell line used in this study on endothelial cells in vitro or for HSPCs in vivo. Selectins were in fact recently described to be diffusive in the apical domain of endothelial cells and to not interact with the submembrane cytoskeleton even transiently. The glycocalyx of the endothelium may also impact the avidity of these interactions: Rolling of cells on the endothelium is indeed a complex process. While it is appreciated that model systems are required to test certain hypotheses, to make conclusions about this behaviour requires some applications to settings that at least approximate these complexities. One can envision 'tethers' forming in the absence of the elasticity of the endothelial cytoskeleton and the elastic potential energy that occurs when glycoproteins are compressed during rolling. These local molecular forces would be predicted to weaken the attachment force of the CD44- or PSGL-1-selectin interactions, perhaps interfering with tethers and slings altogether.

There are also a number of aspects related to the single molecule detection and tracking in this study that are left uncontrolled. To definitively say one is detecting single molecules, for example, a dilution series of the Fab fragments used should be done until a concentration of events mimics that of monodispersed fragments as measured by photobleaching and quantized fluorescence measurements. The authors have not provided these controls and instead indicate that "4-8 fluorophores" per Fab is expected. This can be shown empirically. Once the authors have confidence they are indeed looking at individual Fabs and therefore individual molecules, the authors must determine if they are functionally blocking the binding of the receptors to the E-selectin. If they do, the authors would need to qualify their observations as bystander events that reflect the diffusion of non-engaged receptors.

Importantly, imaging at 15 Hz would not allow the authors to monitor the diffusion of single molecules in three dimensional space and they do not adjust the focus or reconstruct diffusion in 3 dimensions. Yet, they are comparing the diffusive behaviour of their Fabs in tethers (tubes) versus 2-dimensional contact points with the planar ligand. Given the small area of the contact point versus that of the tether, the trajectory of the molecules by definition would become confined more readily.

Given the lack of controls, internal inconsistencies with the writing, and the premise of the study as not being established in physiological settings, this work should be substantially revised before publication here or elsewhere.

Reviewer #2 (Remarks to the Author):

The manuscript by Al Alwan et al. describes and studies the tethering and rolling behaviour of leukemic cells by using a microfluidic platform that mimics the endothelial surface under flow conditions. The authors are able to visualize selectin ligands CD44- and PSGL-1-rich tethers and slings that emanate from the cell's surface and provide evidence that these structures originate

from microvilli where CD44 is pre-clustered in cholesterol-rich departments. Furthermore, they observe increased mobility of selection ligands in tethers and slings, potentially mediated by the detachment of the actin cytoskeleton in these structures. The authors hypothesise that this allows selection ligands to efficiently 'find' their binding partners and thus allow stable rolling.

In general, the manuscript is well written, the data are well described and the arguments the authors make are mostly laid out well. It strongly builds on the groups 2018 Science Advances manuscript but clearly presents new data and is definitely suitable for publication in this journal. However, I have 2 major points that should be clarified and if necessary/possible supported with some experiments.

1. The authors use whole antibodies instead of Fab fragments in their experiments. Supplementary Note 1 shows that Fab fragments give a similar pattern in IFs. However, for single-molecule imaging Fab fragments are pretty much standard so I would like to see a control experiment that shows that Fab fragments produce similar results as seen in Figure 7.

2. The conclusion that detachment of the actin cytoskeleton and thus increased mobility of selectin ligands leads to enhanced ligand-receptor interaction and stable rolling is contradicting earlier studies. E.g. Snapp et al. (PMID: 12036880) showed that the actin-binding cytoplasmic domain of PSGL-1 is required for stable rolling. How do the authors reconcile this with their findings? Given the different experimental systems it might be helpful if the authors address this in their own system (e.g. by expressing CD44/PSGL-1 mutant forms).

I also remember that there is some literature that suggests that only CD44 outside of lipid rafts connects to ERM proteins and the cytoskeleton. M β CD-treatment thus might lead to an increase in ERM-bound CD44, which would strengthen the authors arguments.

Minor point:

- The text often jumps back and forth between Figures. If possible, this should be streamlined to make it easier to read.

Point-by-point response to the reviewers' comments

Reviewer 1

In the manuscript by Bader Al Alwan et al., the authors employ a microfluidic device equipped with ligand-coated chambers to measure the membrane dynamics of rolling cells and, in particular, the diffusion of adhesion receptors (CD44 and PSGL-1). An understanding for the organization and diffusion of receptors that facilitate rolling in the vasculature is indeed incomplete, yet it has long been appreciated that the clustering and the consequential avidity of these interactions is essential. In principle, therefore, this study would pique the interest of a wide-segment of vascular biologists and cell biologists alike. Moreover, single molecule tracking is rarely, if ever done in heterocellular cell-cell contacts or in the context of rolling adhesion biology.

Thank you very much for the positive comment.

Unfortunately, the authors have opted to use planar, rigid arrays of recombinant E-selectin to coat the plastic chambers attached to their microfluidic devices rather than growing normal endothelial monolayers in which the adhesion ligands/receptors can form meta-stable clusters and interact with the submembrane cytoskeleton. While the authors can mimic an appropriate absolute concentration of ligand in their experiments, they cannot control the orientation, diffusion/fluidity, nor the rigidity of the E-selectin. Therefore, it remains unclear if the phenomenon described inclusive of the tethers and slings would also be observed for the KG1a cell line used in this study on endothelial cells in vitro or for HSPCs in vivo. Selectins were in fact recently described to be diffusive in the apical domain of endothelial cells and to not interact with the submembrane cytoskeleton even transiently. The glycocalyx of the endothelium may also impact the avidity of these interactions: Rolling of cells on the endothelium is indeed a complex process. While it is appreciated that model systems are required to test certain hypotheses, to make conclusions about this behaviour requires some applications to settings that at least approximate these complexities. One can envision 'tethers' forming in the absence of the elasticity of the endothelial cytoskeleton and the elastic potential energy that occurs when glycoproteins are compressed during rolling. These local molecular forces would be predicted to weaken the attachment force of the CD44- or PSGL-1-selectin interactions, perhaps interfering with tethers and slings altogether.

Thank you for raising the important issue. The formation of tethers and slings has been observed in a similar system (i.e., neutrophils rolling over P-selectin) using both in vitro microfluidic-based rolling assay on a surface coated by recombinant P-selectin and in vivo intravital imaging,¹ which

strongly suggests the key role played by the tethers/slings in the initial step of the homing. Nevertheless, we agree with the reviewer that we cannot rule out the possibility that the elastic properties of the cells, the diffusional motion of E-selectin on the cells, and other factors may affect the formation of the tethers and slings and their contribution to cell rolling.

In order to address this question, we conducted in vitro cell rolling experiment of KG1a cells over transfected Chinese hamster ovary (CHO) cells expressing E-selectin (CHO-E-sel), a frequently used model system to characterize the rolling behavior of the cells.^{2, 3, 4, 5, 6, 7} CHO-E-sel was cultured in a microfluidic chamber at 37 degree C in the presence of controlled 5% CO₂. The expression of E-selectin was confirmed by immunofluorescence imaging. After confirming the viability of CHO-E-sel in the fluidic chamber, we conducted the cell rolling assay by perfusing the suspension of the KG1a cells into the chamber using a syringe pump at a shear stress of 2 dyne cm⁻². The rolling behavior and the tether/sling formation of the KG1a cells were captured using bright-field and epi-fluorescence microscopy in a way similar to the rolling assay on the surface of the rh E-selectin.

The time-lapse bright field microscopy images showed that the KG1a cells rolled over CHO-E-sel (Figure 1). Next, we perfused the immunostained KG1a cells into the fluidic chamber and captured the rolling KG1a cell using both bright-field and epi-fluorescence microscopy (Figure 2). The bright-field image showed that the KG1a cell rolled over CHO-E-sel (Figure 2 top) while the epi-fluorescence image clearly captured the formation of a tether during the KG1a cell rolling over CHO-E-sel (Figure 2 bottom). Importantly, the time-lapse epi-fluorescence images of the KG1a cell over CHO-E-sel exhibited the formation of tethers and slings and the conversion of the tether to sling (Figure 3). These findings strongly suggest that the formation of the tethers and slings and their dynamic behavior observed in the cell rolling assay that was conducted using the rh E-selectin are highly relevant to the rolling behavior of the cell under physiological conditions.

While our preliminary results indicate that single-molecule analysis of the selectin ligands during the KG1a cell rolling over CHO-E-sel could be possible after the proper optimization of the experimental conditions, detailed characterization of the cell rolling behavior over CHO-E-sel is beyond the scope of this study. We will publish these observations with additional discussion about the contribution of the cellular architecture to the cell rolling behavior elsewhere in near future.

[FIGURE 1 AND 2 REDACTED]

[FIGURE 3 REDACTED]

There are also a number of aspects related to the single molecule detection and tracking in this study that are left uncontrolled. To definitively say one is detecting single molecules, for example, a dilution series of the Fab fragments used should be done until a concentration of events mimics that of monodispersed fragments as measured by photobleaching and quantized fluorescence measurements. The authors have not provided these controls and instead indicate that “4-8 fluorophores” per Fab is expected. This can be shown empirically.

We experimentally determined the average number of fluorophores conjugated to the antibodies (i.e., density of labeling (DOL)). DOL was calculated using the following equations,

$$C = \frac{[A_{280} - (A_{\text{dye}} - CF_{280})]}{\epsilon_{\text{AB}}}$$

$$DOL = \frac{A_{\text{dye}}}{\epsilon_{\text{dye}} \times C}$$

where C and ϵ_{AB} are the concentration and the molar extinction coefficient of the antibody at 280 nm, respectively. A_{280} and A_{dye} denote the absorbance of the dye-antibody conjugate at 280 nm and at the peak absorption wavelength for the respective dye, respectively. ϵ_{dye} and CF_{280} are the molar extinction coefficient of the conjugated dye and a correction factor for the fluorophore's contribution to the absorbance at 280 nm. According to the manufacturer of the dyes used in this study, ϵ_{dye} and CF_{280} are $\epsilon_{\text{dye}} = 71,000$ and $CF_{280} = 0.11$ for Alexa Fluor 488, $\epsilon_{\text{dye}} = 150,000$ and $CF_{280} = 0.08$ for Alexa Fluor 555, and $\epsilon_{\text{dye}} = 239,000$ and $CF_{280} = 0.03$ for Alexa Fluor 647. ϵ_{AB} of the whole antibody and the Fab fragment are 203000 and $75000 \text{ M}^{-1} \text{ cm}^{-1}$, respectively. Therefore, the DOL for AF488-anti-PSGL-1 and AF647-anti-CD44 antibodies displayed in Figure 4 top and middle are estimated to be DOL = 4.1 and 6.8, respectively. DOL for AF555-anti-PSGL-1 antibody (Fab fragment) is estimated to be DOL = 0.46 (Figure 4 bottom). We experimentally determined DOL in each conjugation reaction because DOL in each reaction was not constant. DOL for the whole antibodies was in the range of DOL = 4-9.

After the conjugation reaction, we experimentally determined the fluorescence brightness obtained from single dye-conjugated anti-PSGL-1 and anti-CD44 antibodies (using the dye-conjugated antibodies mentioned above, Figure 4). The dye-conjugated anti-PSGL-1 antibody deposited on a glass surface at the high concentration ($1 \mu\text{g ml}^{-1}$ in Figure 4) showed bright and uniform fluorescence. On the other hand, at lower concentrations (in particular at 0.005 and $0.01 \mu\text{g ml}^{-1}$ concentration in Figure 4), spatially isolated diffraction-limited fluorescence spots were observed. This concentration dependence strongly suggested that each fluorescence spot appeared in these images is a single dye-conjugated anti-PSGL-1 antibody. We experimentally determined the average fluorescence brightness of the single dye-conjugated anti-PSGL-1 antibody using these images. We determined the average fluorescence brightness of the single dye-conjugated anti-CD44 antibody in a similar way. We note that the average fluorescence brightness of the single dye-conjugated anti-PSGL-1 and anti-CD44 antibodies was experimentally determined for each batch of the dye-conjugated antibodies because DOL in each reaction was not constant. We also note that the presence of multiple fluorophores (on average 4-9 dyes per antibody) resulted in brighter fluorescence signal from the immunostained cells,

which allowed us to quantitatively characterize the spatiotemporal dynamics of the selectin ligands (e.g., estimation of the number of the molecules in each spot along the tethers and slings).

We added the above data and discussion to the Supplementary Information of the revised manuscript.

Supplementary Information (page 49): “Supplementary Note 11. **Density of labeling and fluorescence brightness of the antibodies.** We experimentally determined the average number of fluorophores conjugated to the antibodies (i.e., density of labeling (DOL)). DOL was calculated using the following equations,

$$C = \frac{[A_{280} - (A_{\text{dye}} - CF_{280})]}{\epsilon_{\text{AB}}}$$

$$DOL = \frac{A_{\text{dye}}}{\epsilon_{\text{dye}} \times C}$$

where C and ϵ_{AB} are the concentration and the molar extinction coefficient of the antibody at 280 nm, respectively. A_{280} and A_{dye} denote the absorbance of the dye-antibody conjugate at 280 nm and at the peak absorption wavelength for the respective dye, respectively. ϵ_{dye} and CF_{280} are the molar extinction coefficient of the conjugated dye and a correction factor for the fluorophore’s contribution to the absorbance at 280 nm. According to the manufacturer of the dyes used in this study, ϵ_{dye} and CF_{280} are $\epsilon_{\text{dye}} = 71,000$ and $CF_{280} = 0.11$ for Alexa Fluor 488, $\epsilon_{\text{dye}} = 150,000$ and $CF_{280} = 0.08$ for Alexa Fluor 555, and $\epsilon_{\text{dye}} = 239,000$ and $CF_{280} = 0.03$ for Alexa Fluor 647. ϵ_{AB} of the whole antibody and the Fab fragment are 203000 and 75000 $\text{M}^{-1} \text{cm}^{-1}$, respectively. Therefore, the DOL for AF488-anti-PSGL-1 and AF647-anti-CD44 antibodies displayed in Supplementary Figure 29 top and middle are estimated to be DOL = 4.1 and 6.8, respectively. DOL for AF555-anti-PSGL-1 antibody (Fab fragment) is estimated to be DOL = 0.46 (Supplementary Figure 29 bottom). We experimentally determined DOL in each conjugation reaction because DOL in each reaction was not constant. DOL for the whole antibodies was in the range of DOL = 4-9.

After the conjugation reaction, we experimentally determined the fluorescence brightness obtained from single dye-conjugated anti-PSGL-1 and anti-CD44 antibodies (using the dye-conjugated antibodies mentioned above, Supplementary Figure 30). The dye-conjugated anti-PSGL-1 antibody deposited on a glass surface at the high concentration ($1 \mu\text{g ml}^{-1}$ in Supplementary Figure 29) showed bright and uniform fluorescence. On the other hand, at lower concentrations (in particular at 0.005 and $0.01 \mu\text{g ml}^{-1}$ concentration in Figure 30), spatially isolated diffraction-limited fluorescence spots were observed. This concentration dependence strongly suggested that each fluorescence spot appeared in these images is a single dye-conjugated anti-PSGL-1 antibody. We experimentally determined the average fluorescence brightness of the single dye-conjugated anti-PSGL-1 antibody using these images. We determined the average fluorescence brightness of the single dye-conjugated anti-CD44 antibody in a similar way. We note that the average fluorescence brightness of the single dye-conjugated anti-PSGL-1 and anti-CD44 antibodies was experimentally determined for each batch of the dye-conjugated antibodies because DOL in each reaction was not constant. We also note that the presence of multiple fluorophores (on average 4-9 dyes per antibody) resulted in brighter fluorescence signal from the immunostained cells, which allowed us to quantitatively characterize the spatiotemporal

dynamics of the selectin ligands (e.g., estimation of the number of the molecules in each spot along the tethers and slings).”

Figure 4 (Supplementary Figure 34 and 35 in the revised manuscript). (a) Absorption spectra of Alexa Fluor-conjugated antibodies. (b) Fluorescence images of the Alexa-Fluor-647-conjugated anti-PSGL-1 antibody deposited on cover slips. The concentrations of the anti-PSGL-1 antibody in the solutions for the deposition are indicated in each image. The top and bottom images are the same fluorescence images with different image contrasts, which are indicated in the color bars displayed in the right.

Once the authors have confidence they are indeed looking at individual Fabs and therefore individual molecules, the authors must determine if they are functionally blocking the binding of the receptors to the E-selectin. If they do, the authors would need to qualify their observations as bystander events that reflect the diffusion of non-engaged receptors.

To the best of our knowledge and through experiments done in our lab and others, antibodies against glycoprotein ligands, such as PSGL-1 and CD44, that can block binding to E-selectin have not been described or reported.^{8,9} Indeed real time immunoprecipitation studies performed in our lab have shown that these antibodies do not block the ability of the immunoprecipitated protein to recognize E-selectin as illustrated in Figure 4 of reference 9.⁹ To further illustrate this point that the CD44 or PSGL-1 antibodies used in these studies are not blocking the binding of E-selectin to the E-selectin ligands, we performed a flow cytometry experiment where we either stained KG1a cells with recombinant E-selectin followed by an anti-human IgG Fc antibody conjugated to AF-647 (Blue plot), or pre-treated the cells with an antibody to CD44 (515 clone; green) or PSGL-1 (KPL1 clone; red) followed by staining with recombinant E-selectin (Figure 5). As is evident in the graph and much like previous data from our lab, we did not observe a decrease in the ability of the cells to bind E-selectin if the red and green plots are compared to the blue plot.

We added the above data and discussion to the main text and Supplementary Information of the revised manuscript.

Main text (page 6): “We also confirmed that the immunostaining of PSGL-1 and CD44 on the KG1a cells did not affect their binding specificity to E-selectin (Supplementary Figure 6, Supplementary Note 2)”.

Supplementary Information (page 39): “Supplementary Note 2. **Effect of the immunostaining of PSGL-1 and CD44 expressed on the KG1a cells on their binding specificity to E-selectin.** To the best of our knowledge and through experiments done in our lab and others, antibodies against glycoprotein ligands, such as PSGL-1 and CD44, that can block binding to E-selectin have not been described or reported.^{8,9} Indeed real time immunoprecipitation studies performed in our lab have shown that these antibodies do not block the ability of the immunoprecipitated protein to recognize E-selectin.⁹ To further illustrate this point that the CD44 or PSGL-1 antibodies used in these studies are not blocking the binding of E-selectin to the E-selectin ligands, we performed a flow cytometry experiment where we either stained KG1a cells with recombinant E-selectin followed by an anti-human IgG Fc antibody conjugated to Alexa-Fluor-647 (Blue plot), or pre-treated the cells with an antibody to CD44 (515 clone; green) or PSGL-1 (KPL1 clone; red) followed by staining with recombinant E-selectin (Supplementary Figure 6). As is evident in the graph and much like previous data from our lab, we did not observe a decrease in the ability of the cells to bind E-selectin if the red and green plots are compared to the blue plot.”

Figure 5 (Supplementary Figure 7 in the revised manuscript). Flow cytometric analysis of the binding of E-selectin to the KG1a cells in the presence of an antibody to CD44 or PSGL-1. KG1a cells were either incubated with antibodies to CD44 (green), PSGL-1 (red), or without (cyan). Following a washing step, all populations were then incubated with recombinant E-selectin. Control cells represent secondary antibody to E-selectin.

Importantly, imaging at 15 Hz would not allow the authors to monitor the diffusion of single molecules in three dimensional space and they do not adjust the focus or reconstruct diffusion in 3 dimensions. Yet, they are comparing the diffusive behaviour of their Fabs in tethers (tubes) versus 2-dimensional contact points with the planar ligand. Given the small area of the contact point versus that of the tether, the trajectory of the molecules by definition would become confined more readily.

We captured the diffusion of single PSGL-1 molecules at 30 Hz at a single focal plane (i.e., 2D imaging). In order to avoid the defocusing issue (due to the 3D diffusion), we captured the images of tethers and slings that were placed parallel to the surface (Supplementary Movie 4 and 5). This was easy for the slings since many slings were stretched by a laminar flow and aligned parallel to the surface (Supplementary Movie 5). On the other hand, as the reviewer pointed, it was not possible to capture the diffusion of the single molecules along the tethers that do not have any anchoring points (i.e., tethers attached to the surface only at the tethering point). Thus, we obtained the single molecule diffusion data on the tethers by capturing the images of the tethers that attached to the surface at multiple points (i.e., tethering point and anchoring points, Supplementary Movie 4). This allowed us to visualize the entire diffusional motion without the effect of defocusing. We also would like to add that the means square displacement (MSD) plots obtained for the PSGL-1 molecules diffusing on the tethers and slings strongly suggest the random diffusion, not a confined diffusion (i.e., linear relationship between time lag and MSD,

Figure 7b blue and red plots). If the single-molecule diffusion analysis is affected by the defocusing issue, we should observe a confined motion. Thus, our data rather suggests that the analysis was not affected by defocusing issue. In contrast, the PSGL-1 molecule localized on microvilli of the control cells showed a confined diffusion (Figure 7b green plot). The PSGL-1 molecules localized on microvilli stay at the focus during the entire data acquisition time (as the length of microvilli is less than 1 micrometer), and therefore this data suggest that the diffusion of PSGL-1 on microvilli is indeed confined motion.

We added the above discussion to the Supplementary Information of the revised manuscript.

Supplementary Information (page 45): **“Supplementary Note 9. Effect of the three dimensional diffusion along the tethers and slings on the single-molecule tracking analysis.** We captured the diffusion of single PSGL-1 molecules at 30 Hz at a single focal plane (i.e., 2D imaging). Since the PSGL-1 molecules diffuse along tethers and slings, the diffusion does not always occur in the focal plane of the microscope. In order to avoid the defocusing issue (due to the 3D diffusion), we captured the images of tethers and slings that were placed parallel to the surface (Supplementary Movie 4 and 5). This was easy for the slings since many slings were stretched by a laminar flow and aligned parallel to the surface (Supplementary Movie 5). On the other hand, as the reviewer pointed, it was not possible to capture the diffusion of the single molecules along the tethers that do not have any anchoring points (i.e., tethers attached to the surface only at the tethering point). Thus, we obtained the single molecule diffusion data on the tethers by capturing the images of the tethers that attached to the surface at multiple points (i.e., tethering point and anchoring points, Supplementary Movie 4). This allowed us to visualize the entire diffusional motion without the effect of defocusing. In addition, the means square displacement (MSD) plots obtained for the PSGL-1 molecules diffusing on the tethers and slings strongly suggest the random diffusion, not a confined diffusion (i.e., linear relationship between time lag and MSD, Figure 7b blue and red plots). If the single-molecule diffusion analysis is affected by the defocusing issue, we should observe a confined motion. Thus, our data rather suggests that the analysis was not affected by defocusing issue. In contrast, the PSGL-1 molecule localized on microvilli of the control cells showed a confined diffusion (Figure 7b green plot). The PSGL-1 molecules localized on microvilli stay at the focus during the entire data acquisition time (as the length of microvilli is less than 1 micrometer), and therefore this data suggest that the diffusion of PSGL-1 on microvilli is indeed confined motion.”

Given the lack of controls, internal inconsistencies with the writing, and the premise of the study as not being established in physiological settings, this work should be substantially revised before publication here or elsewhere.

We hope that the reviewer is convinced by our above responses, which include additional data and discussion.

Reviewer 2

The manuscript by Al Alwan et al. describes and studies the tethering and rolling behaviour of leukemic cells by using a microfluidic platform that mimics the endothelial surface under flow conditions. The authors are able to visualize selectin ligands CD44- and PSGL-1-rich tethers and slings that emanate from the cell's surface and provide evidence that these structures originate from microvilli where CD44 is pre-clustered in cholesterol-rich departments. Furthermore, they observe increased mobility of selection ligands in tethers and slings, potentially mediated by the detachment of the actin cytoskeleton in these structures. The authors hypothesise that this allows selection ligands to efficiently 'find' their binding partners and thus allow stable rolling. In general, the manuscript is well written, the data are well described and the arguments the authors make are mostly laid out well. It strongly builds on the groups 2018 Science Advances manuscript but clearly presents new data and is definitely suitable for publication in this journal.

Thank you for the positive comment.

The authors use whole antibodies instead of Fab fragments in their experiments. Supplementary Note 1 shows that Fab fragments give a similar pattern in IFs. However, for single-molecule imaging Fab fragments are pretty much standard so I would like to see a control experiment that shows that Fab fragments produce similar results as seen in Figure 7.

Thank you for raising the important issue. According to the suggestion of the reviewer, we conducted single-molecule tracking experiments of the PSGL-1 molecules on the tethers and slings using Fab fragments conjugated to Atto532 dyes. We note that we used Atto532 dye in this experiment because this dye showed brighter fluorescence compared with the Alexa Fluor dyes we used in the experiments reported in the original manuscript. Since the labeling density of the Fab fragments (DOL= 1.5) is much lower than that of the whole antibodies, it was essential to use the dye with very bright fluorescence to capture single molecule images with a reasonable signal-to-background ratio.

As we have already reported in the original manuscript (Supplementary Note Figure 1), we observed a discrete spatial pattern of the PSGL-1 molecules on the tethers and slings (Figure 6a). Next, we conducted single-molecule tracking analysis of the obtained diffusion trajectories. The mean square displacement vs. time lag plots obtained for the PSGL-1 molecules on the tethers and slings were similar to those obtained using the whole antibodies (Figure 6b). Furthermore, the frequency histograms of the diffusion coefficient obtained by the mean square displacement analysis agree well with the original data obtained using the whole antibodies (Figure 6c). We note that the frequency histograms obtained using the Fab fragments are slightly broader compared with those obtained using the whole antibodies. This is mainly due to the less bright fluorescence obtained using the Fab fragments that resulted in less accurate localization and tracking of the molecules. Overall, our results suggest that the single-molecule tracking analysis could be conducted using the whole antibodies without introducing associated artifacts.

We added the above data and discussion to the Supplementary Information of the revised manuscript.

Supplementary Information (page 43): “Supplementary Note 7. **Effect of the immunostaining of PSGL-1 on its single-molecule tracking analysis.** In order to evaluate the effect of the immunostaining of PSGL-1 on its single-molecule tracking analysis, we conducted single-molecule tracking experiments of the PSGL-1 molecules on the tethers and slings using Fab fragments conjugated to Atto532 dyes (Supplementary Figure 25a). The mean square displacement vs. time lag plots obtained for the PSGL-1 molecules on the tethers and slings were similar to those obtained using the whole antibodies (Supplementary Figure 25b). Furthermore, the frequency histograms of the diffusion coefficient obtained by the mean square displacement analysis agree well with the data obtained using the whole antibodies (Supplementary Figure 25c). We note that the frequency histograms obtained using the Fab fragments are slightly broader compared with those obtained using the whole antibodies. This is mainly due to the less bright fluorescence obtained using the Fab fragments that resulted in less accurate localization and tracking of the molecules. Overall, our results suggest that the single-molecule tracking analysis could be conducted using the whole antibodies without introducing associated artifacts.”

Figure 6 (Supplementary Figure 28 in the revised manuscript). (a) Single-molecule fluorescence images of PSGL-1 (immunostained by Atto532-conjugated anti-PSGL-1 Fab) on the sling of the KG1a cell formed during cell rolling over E-selectin at a shear stress of 2 dyne cm^{-2} (0.2 Pa). An example of the single-molecule diffusion trajectories obtained from the time-lapse fluorescence images is shown by the yellow line. (b) Mean square displacement (MSD) versus time lag plots obtained for the PSGL-1 molecules diffusing on the tethers (red open circles) and slings (blue open circles) captured by using the Atto532-conjugated anti-PSGL-1 Fab. The MSD versus time lag plots obtained using the Alexa-Fluor-555-conjugated anti-PSGL-1 whole antibody are displayed as a reference (red circles, blue circles, and green circles for the PSGL-1 molecules on

tethers, slings, and microvilli, respectively). (c) Frequency histograms of the diffusion coefficient of the PSGL-1 molecules on the tethers (top, red) and slings (bottom, blue) of the KG1a cells formed during cell rolling over E-selectin at a shear stress of 2 dyne cm⁻² (0.2 Pa). The diffusion coefficients were calculated by fitting the MSD plots obtained from the individual diffusion trajectories to Eq. 4. The solid lines show Gaussian fittings. The diffusion data obtained using the Alexa-Fluor-555-conjugated anti-PSGL-1 whole antibody are displayed as a reference (black bars and dashed lines for the frequency histograms and Gaussian fittings, respectively).

The conclusion that detachment of the actin cytoskeleton and thus increased mobility of selectin ligands leads to enhanced ligand-receptor interaction and stable rolling is contradicting earlier studies. E.g. Snapp et al. (PMID: 12036880) showed that the actin-binding cytoplasmic domain of PSGL-1 is required for stable rolling. How do the authors reconcile this with their findings?

There are many questions that remain to be answered here. We believe that our results are not in conflict with this data⁴ as we are focused on understanding if differences exist between tethers and slings and other mechanisms of rolling with an emphasis on E-selectin at least initially (reference 4 focused on P-selectin which may or may not pose some differences). Our data supports that microvilli are important since when we lose them (either by M β CD treatment or through the knockdown of CD34), we reduce rolling substantially. The structure of microvilli is partly dependent on bundled actin and ERM proteins. Many studies have discussed the localization of PSGL-1 and CD44 on the tips of microvilli on resting cells and describe how this is critical for rolling on selectins with an emphasis on P-selectin.^{4, 10, 11, 12, 13, 14} Previous work from our lab, focused on E-selectin, clearly shows that the architecture at the surface of the cell changes while rolling and suggests that not all extended structures lead to tethers and slings.¹⁵ Indeed, in that manuscript, using super resolution imaging, we saw that actin was colocalized with CD44 and PSGL-1 in most of the extended structures and this was evident following rolling (see B in below figure from reference 15). In the current manuscript under review here, we looked more closely at tethers and slings and found that there is a separation of the actin cytoskeleton from the tethers and slings and this was consistent with previous force spectroscopy works.^{16, 17, 18} Based on these studies, we suspect that not all of these extended structures (as seen in reference 15 and below) result in tethers/slugs but that this restructuring of the microvilli into these extended actin-filled structures is a pre-requisite to the formation of tethers and slugs that detached from actin. Why this is the case could possibly be due to the fact that a certain amount of bound PSGL-1 to actin is actually necessary for stable rolling and to keep the integrity of the cells structure intact but other PSGL-1 molecules may be able to detach and actually become the anchoring points for tethers and slugs. This is currently under study in our lab and is the focus of future work. We added a discussion of this point in the revised manuscript.

Main text (page 17): “Previous work from our lab using micorfluidics-based super-resolution fluorescence microscopy clearly shows that rolling causes significant reorganization of the clustering behavior of CD44 and PSGL-1, from patchy to elongated network-like structures all

over the surface of the cell.¹⁵ Moreover, in these elongated structures CD44 and PSGL-1 colocalized with actin which supports previous studies implicating the importance of PSGL-1 binding to actin for stable rolling.⁴ Despite the numerous elongated structures we saw,¹⁵ here we observed that only a few of these lead to tethers and slings and thus detach from the actin cytoskeleton to allow the motional freedom of the selectin ligands. It is possible that the restructuring of the microvilli into these elongated actin-filled structures is a pre-requisite to the formation of tethers and slings that detach from actin. Thus, we suspect that both the elongated structures and those that lead to tethers and slings contribute to slow and stable rolling of cells.”

Legend of reference 15: Two-color SR images of CD44 and actin on KG1a cells before and after rolling on E-selectin. (A) SR image of CD44 (cyan) and actin cytoskeleton (yellow) on a KG1a cell that was fixed and immunolabeled by AF-647 (CD44) and labeled by AF-488 dye-conjugated phalloidin (actin). The insets show enlarged views of the yellow regions. (B) SR image of CD44 (cyan) and actin cytoskeleton (yellow) on a KG1a cell that was fixed and fluorescently labeled in the microfluidic chamber **after the rolling** of the cell on E-selectin. The inset shows enlarged view of the yellow region.

Given the different experimental systems it might be helpful if the authors address this in their own system (e.g. by expressing CD44/PSGL-1 mutant forms).

This is a very relevant question but we feel that this is beyond the scope of this manuscript. It is currently the focus of follow-up studies in our lab that are trying to reconcile the formation of microvilli from elongated structures that form after rolling from tethers and slings and the role of ERM proteins (using functional assay as knock-down study) and other relevant signaling molecules involved in actin reorganization and function.

I also remember that there is some literature that suggests that only CD44 outside of lipid rafts connects to ERM proteins and the cytoskeleton. M β CD-treatment thus might lead to an increase in ERM-bound CD44, which would strengthen the authors arguments.

This is a very interesting and relevant point and requires further study to dissect it. We are planning experiments that look more closely at imaging components of lipid rafts, CD44, ERM and actin before and after rolling (as reference 15) as well as real time during rolling (as in the current manuscript). We can also use antibodies to the intracellular domain of CD44 to further observe its interactions with ERM and actin and its effect following rolling. We are very interested in deciphering the differences in the architecture and composition of the tethers and slings compared to the extended structures that result from rolling. Moreover, we are looking into how the different glycoforms of CD44 are distributed on the membrane. We have shown previously that various glycoforms of CD44 are found on the surface of KG1a cells.⁹ It has also been shown previously that O-glycosylation of selectin ligands directs them to lipid rafts.¹³ Therefore, it is likely that the glycosylation on the selectin ligands, i.e. CD44, could help direct it to different membrane domains and thus also dictate it's movement into microvilli and/or tethers and slings. All these studies are fascinating but we feel are not the focus of the current manuscript.

Minor point: The text often jumps back and forth between Figures. If possible, this should be streamlined to make it easier to read.

We reorganized the manuscript, which including 1) moved all the Supplementary Methods to the Methods section of the main text and 2) moved all the Supplementary Note Figures to the Supplementary Figures. We hope that the readability of the paper has been improved by these modifications.

1. Sundd P, *et al.* 'Slings' enable neutrophil rolling at high shear. *Nature* **488**, 399-403 (2012).
2. Moore KL, *et al.* P-selectin glycoprotein ligand-1 mediates rolling of human neutrophils on P-selectin. *J Cell Biol* **128**, 661-671 (1995).
3. Edwards BS, Curry MS, Tsuji K, Brown D, Larson RS, Sklar LA. Expression of P-selectin at low site density promotes selective attachment of eosinophils over neutrophils. *J Immunol* **165**, 404-410 (2000).

4. Snapp KR, Heitzig CE, Kansas GS. Attachment of the PSGL-1 cytoplasmic domain to the actin cytoskeleton is essential for leukocyte rolling on P-selectin. *Blood* **99**, 4494-4502 (2002).
5. Setiadi H, McEver RP. Signal-dependent distribution of cell surface P-selectin in clathrin-coated pits affects leukocyte rolling under flow. *J Cell Biol* **163**, 1385-1395 (2003).
6. Setiadi H, McEver RP. Clustering endothelial E-selectin in clathrin-coated pits and lipid rafts enhances leukocyte adhesion under flow. *Blood* **111**, 1989-1998 (2008).
7. AbuSamra DB, *et al.* Not just a marker: CD34 on human hematopoietic stem/progenitor cells dominates vascular selectin binding along with CD44. *Blood Adv* **1**, 2799-2816 (2017).
8. Merzaban JS, *et al.* Analysis of glycoprotein E-selectin ligands on human and mouse marrow cells enriched for hematopoietic stem/progenitor cells. *Blood* **118**, 1774-1783 (2011).
9. AbuSamra DB, Al-Kilani A, Hamdan SM, Sakashita K, Gadhoum SZ, Merzaban JS. Quantitative characterization of E-selectin interaction with native CD44 and P-selectin glycoprotein ligand-1 (PSGL-1) using a real time immunoprecipitation-based binding assay. *J Biol Chem* **290**, 21213-21230 (2015).
10. Bruehl RE, *et al.* Leukocyte activation induces surface redistribution of P-selectin glycoprotein ligand-1. *J Leukoc Biol* **61**, 489-499 (1997).
11. Buscher K, *et al.* The Transmembrane Domains of L-selectin and CD44 Regulate Receptor Cell Surface Positioning and Leukocyte Adhesion under Flow. *J Biol Chem* **285**, 13490-13497 (2010).
12. Miner JJ, *et al.* Separable requirements for cytoplasmic domain of PSGL-1 in leukocyte rolling and signaling under flow. *Blood* **112**, 2035-2045 (2008).
13. Shao BJ, *et al.* O-glycans direct selectin ligands to lipid rafts on leukocytes. *Proc Natl Acad Sci U S A* **112**, 8661-8666 (2015).

14. von Andrian UH, Hasslen SR, Nelson RD, Erlandsen SL, Butcher EC. A central role for microvillous receptor presentation in leukocyte adhesion under flow. *Cell* **82**, 989-999 (1995).
15. AbuZineh K, Joudeh LI, Al Alwan B, Hamdan SM, Merzaban JS, Habuchi S. Microfluidics-based super-resolution microscopy enables nanoscopic characterization of blood stem cell rolling. *Sci Adv* **4**, eaat5304 (2018).
16. Evans E, Heinrich V, Leung A, Kinoshita K. Nano- to microscale dynamics of P-selectin detachment from leukocyte interfaces. I. Membrane separation from the cytoskeleton. *Biophys J* **88**, 2288-2298 (2005).
17. Heinrich V, Leung A, Evans E. Nano- to microscale dynamics of P-selectin detachment from leukocyte interfaces. II. Tether flow terminated by P-selectin dissociation from PSGL-1. *Biophys J* **88**, 2299-2308 (2005).
18. Shao JY, Ting-Beall HP, Hochmuth RM. Static and dynamic lengths of neutrophil microvilli. *Proc Natl Acad Sci U S A* **95**, 6797-6802 (1998).

REVIEWERS' COMMENTS:

Reviewer #1 (Remarks to the Author):

Major point

The notion that tethers and slings are structures devoid of F-actin that confine surface receptors (to within the afforded surface area of the structure) does not strike me as a conceptual advance.

Minor point

Throughout the figure legends, it is indicated that "antibodies" are used to label CD44 and selectin when I believe the authors have used Fab fragments rather than whole antibodies. Would the authors please indicate in each case if they have used Fabs?

Reviewer #2 (Remarks to the Author):

Dear Authors and Editors,

I'm happy with the changes that have been made to the manuscript and the additional experiments that have been conducted to answer my questions. Therefore, I think it is now ready for publication.